# MULTI-VECTOR RETRIEVAL AS SPARSE ALIGNMENT

## ABSTRACT

Multi-vector retrieval models improve over single-vector dual encoders on many information retrieval tasks. In this paper, we cast the multi-vector retrieval problem as *sparse alignment* between query and document tokens. We propose ALIGNER, a novel multi-vector retrieval model that learns sparsified pairwise alignments between query and document tokens (e.g. '*dog*' vs. '*puppy*') and per-token unary saliences reflecting their relative importance for retrieval. We show that controlling the sparsity of pairwise token alignments often brings significant performance gains. While most factoid questions focusing on a specific part of a document require a smaller number of alignments, others requiring a broader understanding of a document favor a larger number of alignments. Unary saliences, on the other hand, decide whether a token ever needs to be aligned with others for retrieval (e.g. '*kind*' from '*what kind of currency is used in new zealand*'). With sparsified unary saliences, we are able to prune a large number of query and document token vectors and improve the efficiency of multi-vector retrieval. We learn the sparse unary saliences with entropy-regularized linear programming, which outperforms other methods to achieve sparsity. In a zero-shot setting, ALIGNER scores 51.1 points nDCG@10, achieving a new retriever-only state-of-the-art on 13 tasks in the BEIR benchmark. In addition, adapting pairwise alignments with a few examples ($\leq 8$) further improves the performance up to 15.7 points nDCG@10 for argument retrieval tasks. The unary saliences of ALIGNER helps us to keep only 20% of the document token representations with minimal performance loss. We further show that our model often produces interpretable alignments and significantly improves its performance when initialized from larger language models.

## 1 INTRODUCTION

Neural information retrieval (IR) has become a promising research direction for improving traditional IR systems. The most-commonly adopted approach called the dual encoder operates by representing every query and document as a single dense vector. Given sufficient annotations, dual encoders directly learn task-driven similarity between vectors, and often surpass traditional IR systems on complex tasks such as question answering (Lee et al., 2019; Karpukhin et al., 2020; Ni et al., 2021). However, these models can struggle to generalize over out-of-domain datasets (Thakur et al., 2021) and/or entity-centric questions (Sciavolino et al., 2021) due to the limited representational capacity of single vectors. As a remedy, multi-vector retrieval models (Khattab & Zaharia, 2020; Luan et al., 2021; Gao et al., 2021) instead use multiple vectors, typically the contextualized token vectors, to represent the text. These models largely improve the model expressiveness, and exhibit much stronger performance and robustness compared to their single-vector counterparts.

Existing multi-vector retrieval models such as ColBERT (Khattab & Zaharia, 2020) computes query-document similarity by selecting the highest scoring document token for each query token and aggregating the scores. This sum-of-max method has two major limitations. First, restricting the selection to a *single* document token can be highly sub-optimal for some retrieval tasks. As we will show in our experiments, the retrieval performance can be improved by more than 16 points nDCG@10 by relaxing this constraint. Second, the method also leads to a large search index and expensive computation. Specifically, the retrieval and storage cost scales linearly with the query and document length, making multi-vector retrieval models an inferior choice for efficiency-demanding applications. We directly tackle these challenges to build faster and more accurate models.

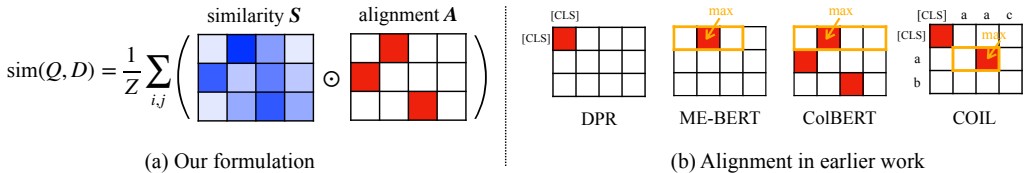

$$\text{sim}(Q,D) = \frac{1}{Z} \sum_{i,j} \left( \text{similarity } \boldsymbol{S} \odot \text{alignment } \boldsymbol{A} \right)$$

(a) Our formulation

(b) Alignment in earlier work

Figure 1: (a) We formulate multi-vector retrieval as token-level sparse alignment; (b) Earlier models can be covered by our formulation as using different alignments.

The representation learning problem of multi-vector retrieval can be formulated as optimizing token-level alignment. Specifically, we use a *sparse alignment matrix* to aggregate token-level similarities, where each element indicates the alignment of a pair of tokens. From this point of view, we are able to formulate different retrieval models in a unified manner (Figure 1) and discern the drawbacks of existing models.

Based on our formulation, we propose ALIGNER, a novel multi-vector retrieval model that consists of pairwise alignment and unary salience. Pairwise alignments form the basis of ALIGNER, where pairs of query and document tokens are sparsely aligned based on their contextual representations. It is discovered that changing the sparsity of alignment can significantly impact the performance on retrieval tasks. For instance, factoid questions often favor a small number of alignments since they often focus on a small part of a document. However, other queries for different tasks (e.g., argument retrieval and fact checking) require a larger number of alignments for a broader understanding of a document. Our findings also support the claim of Dai et al. (2022b) that retrieval tasks with different intents should be modeled differently.

ALIGNER also learns unary saliences, which decides whether each token ever needs to be aligned with any other token for retrieval. This corresponds to masking an entire row or column of the alignment matrix, rather than individual token alignments. To sparsify entire rows or columns, we introduce an algorithm that produces sparse token salience and is end-to-end differentiable based on a novel formulation of entropy-regularized linear programming. Sparsified unary saliences allow us to prune a large number of document and query token representations, making multi-vector retrieval a more efficient and affordable solution.

We evaluate ALIGNER on the BEIR benchmark (Thakur et al., 2021), which covers a diverse set of retrieval tasks in multiple domains.[1] In a zero-shot setting, we show that simply scaling our model achieves the state-of-the-art performance, outperforming prior neural retrievers without contrastive pre-training, model-based hard negative mining, or distillation. By adapting the pairwise alignments with a few examples from the target task — similar to the setup of Dai et al. (2022b) — ALIGNER can be further improved by up to 15.7 points nDCG@10 on argument retrieval tasks. Meanwhile, pruning with our unary saliences can reduce 50% of query tokens for better run-time efficiency and 80% of document tokens for better storage footprint, with less than 1 point decrease of nDCG@10. The pairwise alignments and unary saliences are also highly interpretable so that they often serve as concise rationales for retrieval.

## 2 MULTI-VECTOR RETRIEVAL AS SPARSE ALIGNMENT

Given a query $Q$ and a collection of $N$ documents $\mathscr{C} = \{D^{(1)}, \ldots, D^{(N)}\}$, a key problem in retrieval is how to represent these textual inputs in order to facilitate efficient search. To this end, one approach is lexical retrieval using sparse bag-of-words representation of the text; the other approach is dense retrieval, which this work focuses on. Dense retrieval models learn a parameterized function that encodes the query and documents into query representation $\boldsymbol{q}$ and document representations $\{\boldsymbol{d}^{(1)}, \ldots, \boldsymbol{d}^{(N)}\}$ respectively. Typically, each representation is a single $d$-dimensional vector. For retrieval, the similarity function is often defined as $\text{sim}(Q, D^{(i)}) = \boldsymbol{q}^\top \boldsymbol{d}^{(i)}$, and documents having high similarity scores to the query are retrieved.

---

[1]We will release our model checkpoints to encourage future research.

## 2.1 MULTI-VECTOR RETRIEVAL

Instead of representing each query and document as a single fixed-length vector, multi-vector retrieval represents them with multiple token vectors, mainly to improve the limited capacity of fixed-length representations. Specifically, a query $Q = \{q_1, \ldots, q_n\}$ and a document $D = \{d_1, \ldots, d_m\}$ are encoded into a set of vectors $\{\boldsymbol{q}_1, \ldots, \boldsymbol{q}_n\}$ and $\{\boldsymbol{d}_1, \ldots, \boldsymbol{d}_m\}$. The similarity function between a query and a document is re-defined for multi-vector retrieval. For instance, ColBERT (Khattab & Zaharia, 2020) designs the similarity function as follows:

$$\text{sim}(Q,D) = \sum_{i=1}^{n} \max_{j=1\ldots m} \boldsymbol{q}_i^\top \boldsymbol{d}_j.$$

For retrieval, instead of indexing $N$ document vectors, multi-vector retrieval pre-computes $N \times \bar{m}$ document *token* vectors where $\bar{m}$ is the average length of documents. Then, it retrieves $K$ document token vectors for each query token vector with Maximum Inner-Product Search (MIPS), resulting in $n \times K$ candidate document tokens. The retrieved tokens are used to trace back the original documents (Lee et al., 2021a), often followed by a final refinement stage that scores the similarity $\text{sim}(Q,D)$ with all token representations of each document and the query (Khattab & Zaharia, 2020). We adopt the same practice of ColBERT in our experiments.

## 2.2 SPARSE ALIGNMENT FORMULATION

A key design question for retrieval models is defining the similarity function in a manner that balances model expressiveness and inference cost. To facilitate our discussion, we formalize the similarities used in previous methods into a class of sparse alignment functions. The formulation also leads to a principled extension over existing work, which we will describe in §3.

We begin by defining a *similarity matrix* $\boldsymbol{S} \in \mathbb{R}^{n \times m}$ computed from all pairs of query and document tokens, where $\boldsymbol{S}_{i,j} = \boldsymbol{q}_i^\top \boldsymbol{d}_j$. Then, we use an *alignment matrix* $\boldsymbol{A} \in [0,1]^{n \times m}$ to compute the similarity between $Q$ and $D$ as follows:

$$\text{sim}(Q,D) = \frac{1}{Z} \sum_{i=1}^{n} \sum_{j=1}^{m} \boldsymbol{S}_{i,j} \boldsymbol{A}_{i,j} \tag{1}$$

where $Z$ is a normalization term defined as $Z = \sum_{i,j} \boldsymbol{A}_{i,j}$. The alignment matrix $\boldsymbol{A}$ can be directly derived from $\boldsymbol{S}$ or computed as a function of $Q$ and $D$.

We constrain the alignment matrix $\boldsymbol{A}$ to be *sparsely* activated: $||\boldsymbol{A}||_0 \leq \sigma$ where $||\cdot||_0$ is the number of non-zero elements in a matrix. Sparse activation assumes that only a few query-document token matches are critical for retrieval, inspired by traditional retrieval methods. Indeed, most existing dense retrieval models already enforce the sparse alignment with their own heuristics. Figure 1 illustrates how different models can be described under our formulation:

- **Dense passage retriever** (DPR; Karpukhin et al., 2020) uses a single [CLS] vector to represent each query and document. This is equivalent to setting $A_{1,1} = 1$ and 0 otherwise, resulting in $||\boldsymbol{A}||_0 = 1$.
- **ME-BERT** (Luan et al., 2021) uses the first $k$ document token vectors for multi-vector representations of documents but a single vector for query. The similarity function is $\max_{j=1\ldots k} \boldsymbol{q}_1^\top \boldsymbol{d}_j$, which is equivalent to setting $A_{1,j} = 1$ when $\boldsymbol{S}_{1,j}$ is the maximum within $\boldsymbol{S}_{1,1}$ to $\boldsymbol{S}_{1,k}$, and 0 otherwise. The alignment sparsity is $||\boldsymbol{A}||_0 = 1$.
- **ColBERT** uses the sum-of-max similarity function $\sum_{i=1}^{n} \max_{j=1\ldots m} \boldsymbol{S}_{i,j}$ that is equivalent to setting an alignment matrix to select the maximum element from each row of $\boldsymbol{S}$, i.e., $A_{i,j} = 1$ when $\boldsymbol{S}_{i,j}$ is the maximum within $\boldsymbol{S}_{i,:}$. $||\boldsymbol{A}||_0 = n$ in this case.
- **COIL** (Gao et al., 2021), similar to ColBERT, also selects the maximum element from each row of $\boldsymbol{S}$, but requires a lexical exact match for a selected pair, i.e., $A_{i,j} = 1$ when $\boldsymbol{S}_{i,j}$ is the maximum within $\{\boldsymbol{S}_{i,j'} \mid q_i = d_{j'}\}$. $||\boldsymbol{A}||_0 \leq n$ in this case.

The choice of similarity and sparsity can have a large impact on model capacity and efficiency. For instance, ColBERT is more expressive and robust than DPR (Thakur et al., 2021), but its retrieval and storage costs are much higher. Our work seeks to further advance expressiveness while retaining a strong efficiency. We describe our method in the next section.

Figure 2: ALIGNER factorizes the alignment matrix into pairwise alignments and unary saliences. Pairwise alignment focuses on the alignment of individual token pairs. Unary saliences are determined by per-token salience features.

# 3 ALIGNER

In this section, we present ALIGNER built upon the sparse alignment formulation. ALIGNER factorizes the alignment matrix into pairwise alignment and unary salience:

$$\boldsymbol{A} = \tilde{\boldsymbol{A}} \odot (\boldsymbol{u}^q \otimes \boldsymbol{u}^d) \tag{2}$$

where $\odot$ is the Hadamard product and $\otimes$ is the outer product of two vectors. Pairwise alignment $\tilde{\boldsymbol{A}} \in \mathbb{R}^{n \times m}$ determines which pairs of query and document tokens should be aligned, with the sparsity constraints tailored for downstream tasks (§3.1). Unary salience $\boldsymbol{u}^q \in \mathbb{R}^n$ and $\boldsymbol{u}^d \in \mathbb{R}^m$ are sparse token weights deciding whether a token ever needs to be aligned (§3.2).

The factorization is introduced based on two critical hypotheses. First, the optimal sparsity of alignment can be *task-dependent*. Instead of imposing top-1 constraint as in ColBERT, activating more than one alignments for a query token can enhance retrieval performance for certain tasks. In our analyses for instance, we observe factoid questions that only concern a specific part of a document require a small number of alignments, while some other queries (such as fact checking) require more alignments for a broader understanding of the document. We explore different search spaces of the pairwise alignment matrix $\tilde{\boldsymbol{A}}$ in order to achieve better retrieval performance for each downstream task. Second, alignment is only needed for *very few* tokens. For example, we analyzed 2000 most retrieved documents in our preliminary study, and found only 12.8% document tokens are retrieved by at least one query.[2] Intuitively, tokens that are uninformative do not need to be aligned and stored, corresponding to sparse activation over an entire row or column of $\boldsymbol{A}$. ALIGNER directly learns the row and column sparsity as unary salience, and utilizes them to enhance retrieval efficiency.

## 3.1 ADAPTING PAIRWISE ALIGNMENT

Queries and documents can have varied distributions. For example, a query can be a single entity, a natural question, or a few sentences, and a document can range from a short paragraph to a long article. The search intent also changes from task to task (Dai et al., 2022b). These changes can lead to different optimal alignment strategies. We explore the following sparse alignment variants that go beyond the top-1 strategy commonly adopted in existing work:

- **Top-$k$.** Each query token is aligned with $k$ document tokens with highest similarity scores. Precisely, $\tilde{\boldsymbol{A}}_{i,j} = 1$ when the $j$-th token is within top-$k$ of the row $\boldsymbol{S}_i$. When $k = 1$, it is equivalent to ColBERT.

- **Top-$p$.** This strategy is similar to top-$k$, but instead of aligning each query token with exactly $k$ tokens, it makes the number of alignments proportional to the document length, i.e., each query token aligns with $\max(\lfloor p \cdot m \rfloor, 1)$ tokens where $m$ is the document length and $p \in [0, 1]$ is the alignment ratio.

Despite their simplicity, these variants can indeed enhance retrieval accuracy significantly on tasks such as argument retrieval. More importantly, while it is possible to train separate models for different alignment variants, we are interested in fast test-time adaptation using a single shared model as many important retrieval tasks lack sufficient training data (Thakur et al., 2021). Specifically,

---

[2]The analysis was performed on MS MARCO (Nguyen et al., 2016) using our implementation of ColBERT.

we first train ALIGNER using a fixed alignment strategy such as top-1 in a source domain, and adapt the alignment strategy to each target task without changing the model parameters.[3] We use the following few-shot alignment adaptation method. Given a corpus $\{D^{(1)}, \ldots, D^{(N)}\}$, and a few relevance-annotated query-document pairs from the target task $\{(Q^1, D^1_+), \ldots (Q^K, D^K_+)\}$, we first retrieve candidate documents with the learned token representations, and decide the pairwise alignment strategy based on the ranking performance on the annotated data. This adaptation can be performed efficiently because the alignment only concerns the computation of similarity score (Eq. 1) in the refinement stage. In practice, for some tasks, we are able to find a well-suited alignment strategy and improve the retrieval performance with as few as 8 annotated examples.

## 3.2 LEARNING UNARY SALIENCE

ALIGNER predicts token saliences from their token representations. For brevity, we only present the formulation for document salience, and query salience is defined similarly. Specifically, the salience of the $i$-th document token $u^d_i$ is defined as:

$$u^d_i = \lambda^d_i \cdot f(\boldsymbol{W}^d \boldsymbol{d}_i + b^d) \tag{3}$$

where $\boldsymbol{W}^d$ and $b^d$ are learnable parameters. $f$ is a non-linear activation function and we use ReLU such that salience is always non-negative. $\boldsymbol{\lambda}^d = \{\lambda^d_i\}$ are gating variables to control the overall sparsity of $\boldsymbol{u}^d$, which we will elaborate next.

For the document salience to be meaningful, we enforce salience sparsity as an inductive bias. ALIGNER jointly optimizes sparse salience with other parts of the model. Since tokens with zero salience do not contribute to computing similarity, our model will be encouraged to identify more important tokens in order to retain good retrieval performance. Note that during training we do not have any explicit annotation on which tokens are important. Instead, $\boldsymbol{u}^d$ (and similarly $\boldsymbol{u}^q$) are directly optimized to minimize the training loss, under the sparsity constraint that $\|\boldsymbol{\lambda}^d\|_0 = \lceil \alpha^d \cdot m \rceil$, where $\alpha^d$ is a constant sparsity ratio and $m$ is the document length.

A key question is how we can optimize the unary salience component given the controlled sparsity. We leverage a novel technique called entropy-regularized linear programming to enable end-to-end optimization. Specifically, let $k = \lceil \alpha^d \cdot m \rceil$ denotes the desired sparsity, $s_i = f(\boldsymbol{W}^d \boldsymbol{d}_i + b^d)$ denotes the token score before the sparse gate $\lambda^d_i$ is applied, and $\boldsymbol{s}, \boldsymbol{\lambda}^d \in \mathbb{R}^m$ be the vectors $\{s_i\}$ and $\{\lambda^d_i\}$ respectively. $\boldsymbol{\lambda}^d$ is computed by solving the following optimization problem:

$$\max_{\boldsymbol{\lambda}} \ \boldsymbol{s}^\top \boldsymbol{\lambda} + \varepsilon H(\boldsymbol{\lambda}) \qquad \text{s.t.} \ \ \mathbf{1}^\top \boldsymbol{\lambda} = k, \ \lambda_i \in [0,1], \ \forall i = 1, \ldots, m. \tag{4}$$

where $H(\cdot)$ is the elementwise entropy function[4] and $\varepsilon > 0$ is a small constant. The optimization can be seen as a relaxed top-$k$ operation. Without the entropy term $\varepsilon H(\cdot)$, it becomes an instance of linear programming where the solution $\boldsymbol{\lambda}^d$ is a binary mask indicating the top-$k$ values of $\boldsymbol{s}$, i.e., $\lambda^d_i = 1$ if and only if $s_i$ is one of top-$k$ values in $\boldsymbol{s}$. This top-$k$ optimization is smoothed by adding the small entropy term $\varepsilon H(\cdot)$ and by relaxing $\lambda_i$ from exact binary to $[0,1]$. Given small $\varepsilon$, this still produce a sparse solution $\boldsymbol{\lambda}^d$ and can be solved using simple vector operations. Specifically, let $a \in \mathbb{R}$ and $b_i \in \mathbb{R}$ for $i = 1, \cdots, m$ be auxiliary variables that are initialized to zero. We iteratively update these variables using the following equations:

$$a' = \varepsilon \ln(k) - \varepsilon \ln \left\{ \sum_i \exp \left( \frac{s_i + b_i}{\varepsilon} \right) \right\}, \qquad b'_i = \min(-s_i - a', 0). \tag{5}$$

In practice, it is sufficient to run only a few iterations and the final solution is given by $\lambda_i = \exp(\frac{s_i + b_i + a}{\varepsilon})$. These vector operations are differentiable so $\boldsymbol{\lambda}$ can be end-to-end trained with other parts of our model. The full derivation of this iterative algorithm is given in Appendix A.1.

---

[3]We have also explored a differentiable alignment with sparsity constraints (Appendix A.2), but alignment adaptation is still necessary to achieve good performance on target tasks.

[4]$H(\boldsymbol{\lambda}) = \sum^m_{i=1} -\lambda_i \log \lambda_i$. $\boldsymbol{\lambda}$ is not a probability distribution. $H(\boldsymbol{\lambda})$ is an extension of the entropy function that is applied to any positive vector $\boldsymbol{\lambda}$.

| | Supervision | Hard Negatives | Distillation | Retriever | Per-domain | # Param. |
|---|---|---|---|---|---|---|
| Splade$_{v2}$ | MS MARCO | model-based | ✓ | lexical | | 110M |
| ColBERT$_{v2}$ | MS MARCO | model-based | ✓ | multi-vector | | 110M |
| GTR$_{base / xxl}$ | Pre-train + MS MARCO | fixed | | single-vector | | 110M / 6B |
| PROMPTAGATOR | Few ($\leq 8$) | | | single-vector | ✓ | 137B + 110M |
| ALIGNER$_{base / xxl}$ | MS MARCO + Few* ($\leq 8$) | fixed | | multi-vector | | 110M / 6B |

Table 1: Comparison of different retrieval models. *: optionally used for alignment adaptation.

**Pruning Multi-vector Retrieval**   Despite good retrieval performance, multi-vector retrieval models are notorious for its expensive token-level retrieval (Santhanam et al., 2022). This prevents multi-vector retrievers from being widely adopted in practice. With the learned unary salience, we can prune reduent tokens for multi-vector retrieval. Pruning document tokens reduces the number of vectors in search index, and pruning query tokens reduces the number of searches. Compared to the sparse salience method using L1-norm regularization (Hofstätter et al., 2022), our proposed method offers a direct control over the sparsity. In our experiments, we prune query and document tokens using two pruning ratios $\beta^q$ and $\beta^d$ respectively. For each document, we obtain the token salience using Eq.(3) and only store the top $\beta^d$ percent of tokens in the index. Similarly we select the top $\beta^q$ percent query tokens to perform max inner-product search. Note that we vary these two ratios to control retrieval efficiency, and these ratios can be smaller than the sparsity ratio $\alpha^q$ and $\alpha^d$ which we use as constraints at training time. In the refinement stage, we still use the full model with all token vectors for scoring. With token embedding caching, the computation cost of refinement is relatively small compared to that of retrieval.

## 4   EXPERIMENTS

### 4.1   EXPERIMENTAL SETUP

ALIGNER uses shared transformer encoder initialized from T5 version 1.1 (Raffel et al., 2020). We project token embeddings to 128 dimension and apply L2 normalization. Following GTR (Ni et al., 2021), we finetune ALIGNER on MS MARCO with hard negatives released by RocketQA (Qu et al., 2021). The models are trained with a batch size of 256 for 25k steps, using query sequence length of 64 and document sequence length of 256. We train ALIGNER with top-1 pairwise alignment.[5]

For retrieval, we pre-compute the token encodings of all the documents in the corpus, and use ScaNN (Guo et al., 2020) to index and perform max inner-product search (MIPS). We retrieve 4,000 nearest neighbors for each query token,[6] and return the top-1,000 after the refinement stage. We evaluate ALIGNER on the BEIR benchmark (Thakur et al., 2021) and compare with state-of-the-art retrieval models shown in Table 1. Note that ALIGNER does not rely on contrastive model pre-training (Izacard et al., 2022; Ni et al., 2021), model-based hard negative mining (Santhanam et al., 2021), or distillation (Santhanam et al., 2021). We intentionally decide this simple recipe and focus on studying the impact of pairwise alignment and unary salience.

For few-shot alignment adaptation of ALIGNER (§3.1), we split the test data into multiple folds such that each fold contains 8 examples. Then we find the best alignment strategy that maximizes nDCG@10 on each fold with $k \in \{1, 2, 4, 6, 8\}$ for top-$k$ and $p \in \{0.5\%, 1\%, 1.5\%, 2\%\}$ for top-$p$. Based on the best alignment strategy from each fold, we measure the retrieval performance on the remaining test examples with the best strategy. We report the average ($\pm$ std.) of these test scores where the number of test scores equals the number of folds. The average of few-shot adaptation indicates the expected performance of using few examples to choose the best alignment strategy.

### 4.2   RETRIEVAL ACCURACY

Table 2 shows the document retrieval performance of ALIGNER on both MS MARCO and the BEIR benchmark. For this experiment, we do not prune any query or document tokens with unary saliences, but show their effects in §4.3 instead. ALIGNER$_{base}$ outperforms state-of-the-art sparse and dense retrievers on MS MARCO. It performs slightly worse than ColBERT$_{v2}$, given that we do not use distillation or model-based hard negatives to optimize the in-domain performance.

---

[5]We have trained models with other top-$k$ and top-$p$ pairwise alignments, but the MS MARCO training data favors top-1 alignment (see Appendix A.4 for details).

[6]Unlike ColBERT, ALIGNER does not use pad token embeddings for retrieval. Hence, retrieving 4,000 neighbors per query token results in a similar number of retrieved candidates to ColBERT.

| | Zero-shot | | | | | | | | | Few-shot | | |
|---|---|---|---|---|---|---|---|---|---|---|---|---|
| | BM25 | Splade*_v2 | DPR | GTR_base | GTR_xxl | ColBERT*_v2 | ColBERTer | ALIGNER_base | ALIGNER_xxl | Promptagator | ALIGNER_base | ALIGNER_xxl |
| MS MARCO | 18.7 | 36.8 | 31.1 | 36.6 | 38.8 | 39.7 | 38.7 | 38.8 | **40.3** | - | - | - |
| ArguAna | 31.5 | 47.9 | 41.4 | 51.1 | **54.0** | 46.3 | - | 28.8 | 33.8 | 59.4 | $46.2^{\pm3.6}$ | $47.9^{\pm3.0}$ |
| Touché-2020 | **36.7** | 27.2 | - | 20.5 | 25.6 | 26.3 | - | 34.8 | 34.5 | 34.5 | $49.9^{\pm1.2}$ | $50.2^{\pm1.1}$ |
| FEVER | 75.3 | **78.6** | 58.9 | 66.0 | 74.0 | 78.5 | - | 72.4 | 74.2 | 77.0 | $71.2^{\pm5.9}$ | $73.9^{\pm4.8}$ |
| Climate-FEVER | 21.3 | 23.5 | 17.6 | 24.1 | **26.7** | 17.6 | - | 18.1 | 19.7 | 24.0 | $21.8^{\pm2.2}$ | $22.8^{\pm2.9}$ |
| SciFact | 66.5 | 69.3 | 47.8 | 60.0 | 66.2 | 69.3 | - | 70.4 | **73.1** | 65.0 | $68.8^{\pm2.1}$ | $71.4^{\pm2.2}$ |
| TREC-COVID | 65.6 | 71.0 | 56.1 | 53.9 | 50.1 | 73.8 | - | 68.3 | **75.8** | 75.6 | $73.3^{\pm1.4}$ | $79.3^{\pm3.0}$ |
| NFCorpus | 32.5 | 33.4 | 20.8 | 30.8 | 34.2 | 33.8 | - | 34.0 | **35.2** | 33.4 | $32.9^{\pm2.0}$ | $33.4^{\pm2.0}$ |
| NQ | 32.9 | 52.1 | 39.8 | 49.5 | 56.8 | 56.2 | - | 52.2 | **60.5** | - | $49.0^{\pm4.4}$ | $56.6^{\pm5.1}$ |
| HotpotQA | 60.3 | **68.4** | 37.1 | 53.5 | 59.9 | 66.7 | - | 61.7 | 65.2 | 61.4 | $59.5^{\pm2.9}$ | $63.2^{\pm3.3}$ |
| FiQA | 23.6 | 33.6 | 27.5 | 34.9 | **46.7** | 35.6 | - | 33.4 | 43.5 | 46.2 | $29.8^{\pm4.0}$ | $39.9^{\pm4.5}$ |
| SCIDOCS | 15.8 | 15.8 | 10.8 | 14.9 | 16.1 | 15.4 | - | 14.1 | **17.1** | 18.4 | $14.4^{\pm0.8}$ | $16.3^{\pm1.2}$ |
| DBPedia | 31.3 | 43.5 | 23.6 | 39.2 | 40.8 | 44.6 | - | 41.6 | **45.0** | 38.0 | $40.6^{\pm2.0}$ | $43.5^{\pm2.4}$ |
| Quora | 78.9 | 83.5 | 84.2 | 88.1 | **89.2** | 85.2 | - | 82.3 | 86.0 | - | $82.1^{\pm2.0}$ | $85.3^{\pm2.1}$ |
| **Average** | 44.0 | 49.8 | - | 45.1 | 49.3 | 49.9 | - | 47.1 | **51.1** | - | $49.2^{\pm3.0}$ | $52.6^{\pm3.1}$ |
| – NQ / Quora | 41.9 | 46.6 | - | 40.8 | 44.9 | 46.2 | - | 43.4 | **47.0** | 47.8 | $46.2^{\pm2.5}$ | $49.3^{\pm2.7}$ |

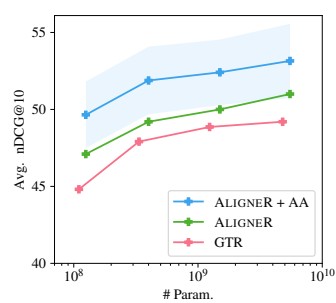

Table 2: Results on MS MARCO (top; MRR@10) and the BEIR benchmark (bottom; nDCG@10). Best zero-shot scores are denoted in boldface. *: trained with distillation. †: uses few examples (≤8) for task-specific adaptation.

Figure 3: Avg. nDCG@10 on BEIR with different sized models. AA: alignment adaptation.

| | Top-$k$ | | | | | | Top-$p$ | | | | |
|---|---|---|---|---|---|---|---|---|---|---|---|
| | 1 | 2 | 4 | 6 | 8 | 16 | 0.5% | 1% | 1.5% | 2% | 5% |
| ArguAna | 28.8 | 24.4 | 18.3 | 14.5 | 11.4 | 5.1 | 33.3 | 45.5 | **48.1** | 46.9 | 32.6 |
| Touché-2020 | 34.8 | 50.0 | **51.1** | 49.3 | 46.0 | 33.7 | 31.3 | 24.2 | 20.3 | 15.8 | 5.2 |
| SciFact | 70.4 | 68.8 | 65.0 | 60.9 | 55.6 | 38.6 | **71.1** | 69.4 | 67.2 | 62.7 | 39.0 |
| TREC-COVID | 68.3 | **74.0** | 73.2 | 67.2 | 61.7 | 41.8 | 66.8 | 64.4 | 56.7 | 46.7 | 30.9 |
| FiQA | **33.4** | 30.8 | 23.8 | 19.5 | 15.5 | 8.43 | **33.4** | 28.5 | 24.6 | 19.0 | 7.7 |
| SCIDOCS | 14.1 | 14.3 | 13.1 | 11.4 | 9.7 | 4.82 | 14.4 | **14.9** | **14.9** | 14.8 | 8.1 |
| DBPedia | **41.6** | 39.4 | 29.6 | 20.2 | 14.2 | 3.94 | **41.6** | 41.7 | 39.9 | 36.6 | 17.0 |
| Average | 41.6 | 43.1 | 39.2 | 36.5 | 30.6 | 19.5 | 41.7 | 41.2 | 38.8 | 34.6 | 20.1 |

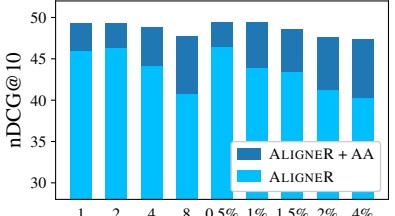

Table 3: nDCG@10 on the BEIR benchmark with different $k$ and $p$ in ALIGNER_base (trained with top-$k = 1$).

Figure 4: Avg. nDCG@10 on BEIR of ALIGNER trained with different alignments (top-$k = 1, 2, 4, 8$ and top-$p = 0.5\%, 1\%, \ldots, 4\%$).

ALIGNER_xxl achieves the strongest results, showing multi-vector retrieval models benefit from large pretrained language models. It also outperforms GTR_xxl on 9 out of 13 BEIR datasets and advances the retriever-only state-of-the-art (ColBERT_v2) by 1.2 points nDCG@10 on average. Figure 3 shows that our multi-vector retriever model scales better than single-vector dual encoder GTR.

**Alignment Adaptation** In the rightmost column of Table 2, we show the effect of adapting pairwise alignment with ALIGNER on the BEIR benchmark. With only 8 examples for finding the proper alignment sparsity, its expected performance reaches 52.6 nDCG@10 on average. Alignment-adapted ALIGNER also benefits from scaling up, and consistently outperforms its non-adapted counterparts, as shown in Figure 3. The gains are further explained in Table 3, where we show individual task's performance under various alignment strategies. Although ALIGNER is trained with top-1 alignment, top-1 is not always the best strategy at inference time. Specifically, for ArguAna, we observe 16 points improvement by adjusting the number of alignments proportional to the document length with $p = 1.5\%$. In general, keeping the sparsity low enough is preferable and supports our hypothesis that pairwise alignments should be sparse.

Figure 4 compares ALIGNER variants trained with other pairwise alignment strategies, including top-$k = 1, 2, 4, 8$ and top-$p = 0.5\%, \ldots, 4\%$. We evaluate their performance with training-time alignment strategy (default) and the optimal alignment strategy selected per dataset (oracle). Among these variants, top-1 and top-0.5% work the best, and increasing $k$ or $p$ hurts performance. Alignment adaptation improves all models, showing that retrieval tasks require different pairwise alignment.

Figure 5 shows the effectiveness of few-shot alignment adaptation—dynamically selecting task-specific alignment strategy based on a few examples. When the default alignment (top-$k$=1) is not optimal, we can identify a good alignment strategy using only 8 examples, which significantly improves model performance on argument retrieval tasks. Using 16 examples further improves the

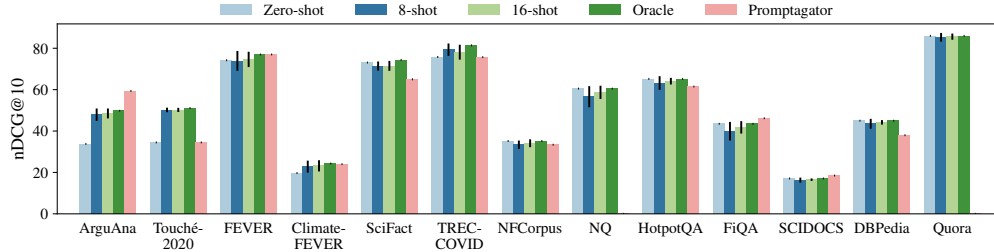

Figure 5: ALIGNER$_{xxl}$ with few-shot alignment adaptation. We report nDCG@10 on BEIR.

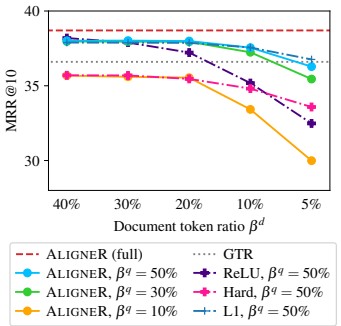

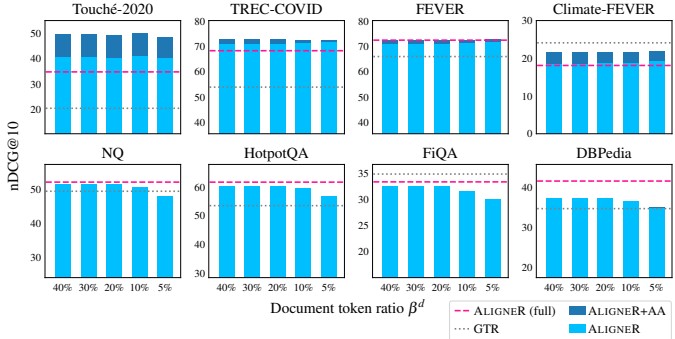

Figure 6: ALIGNER with unary salience on MS MARCO. $\beta^q$ and $\beta^d$ are ratios to prune query and document tokens, respectively.

Figure 7: ALIGNER with unary salience on BEIR. We set query pruning ratio $\beta^q = 50\%$ and vary document pruning ratio $\beta^d$. We omit datasets with small corpora. We also report the performance of ALIGNER with alignment adaptation (AA).

average score and reduces the variance. However, when the default alignment is already optimal (top-$k$=1 is optimal for QA tasks), few-shot alignment adaptation hurts performance due to the variance of our few-shot method. Nevertheless, ALIGNER outperforms Promptagator (Dai et al., 2022b), another few-shot retrieval baseline, in 6 out of 11 datasets.

## 4.3 RETRIEVAL EFFICIENCY

Next we show how ALIGNER's unary salience improves retrieval efficiency. We train ALIGNER$_{base}$ with salience sparsity ratios ($\alpha^q = 50\%, \alpha^d = 40\%$) and $\varepsilon = 0.002$ based on empirical performance. At retrieval time, we prune query and document tokens with ratios $\beta^q$ and $\beta^d$ (§3.2).

Figure 6 shows the ALIGNER performance on MS MARCO with various pruning ratios. When pruned at the same ratio as training ($\beta^q = 50\%, \beta^d = 40\%$), the model performs similarly to a full model (MRR@10 38.1 vs. 38.8). We can further prune tokens by adjusting $\beta^d$ and $\beta^q$. The model achieves 37.3 MRR@10 with $\beta^d = 10\%$, i.e. it remains accurate with only 10% of the original index size. Decreasing the query pruning ratio $\beta^q$ to 30% does not sacrifice performance too much, although deceasing $\beta^q$ to 10% leads to worse performance. Figure 6 also compares ALIGNER's entropy-regularized linear program (Eq. 4) with alternative methods. With just a ReLU gate and no sparsity constraints ('ReLU' in Figure 6), the model retains a good performance when $\beta^d = 40\%$, but degrades for smaller $\beta^d$. Removing the entropy regularization in Eq. 4 leads to simply selecting the hard top-$k$ tokens with the highest predicted salience ('Hard' in Figure 6). The hard top-$k$ solution performs worse for all $\beta^d$. Another method to sparsify salience is adding an L1-norm regularization ('L1' in Figure 6). With a proper coefficient, it performs comparably to our method and slightly better when $\beta^d = 5\%$. Note that our method has the advantage of explicitly controlling the sparsity ratios, without tuning the coefficient of the L1 term.

ALIGNER's salience estimation also generalizes to other retrieval datasets. As shown in Figure 7, pruning with $\beta^d = 20\%$ with $\beta^q = 30\%$ causes minimal performance decrease for a majority of BEIR datasets, which leads to an 80% reduction in the search index size and up to 94% reduction in the computation cost[7]. We even observe performance increase for Touché-2020, as the model can only retrieve salient tokens after pruning. Besides, we show that alignment adaptation can be combined with pruning, resulting in an effective yet efficient retrieval model.

---

[7]Assume brute force search.

| Query | what happens when **stop** **drinking** **alcohol** |
|-------|-----------------------------------------------------|
| Doc. | **alcohol** . symptoms of alcohol **withdrawal** may begin from 4 to 12 hours after you cut down or stop **drinking** , or as long as several days after the last drink , and can last a few days . they can range from mild to life - threatening . 1 mild withdrawal symptoms may include : 2 intense worry . 3 nausea or vomiting . 4 s hak iness . 5 sweat ing . |
| Query | where is the **heart** in the **human** **body** |
| Doc. | **heart** the heart is a **muscular** **organ** in most animals , which pumps blood through the blood vessels of the circul atory system . [ 1 ] blood provides the body with oxygen and nutrients , as well as assists in the removal of metabolic waste s . [ 2 ] in humans , the heart is located between the lungs , in the middle compartment of the chest . [ 3 ] |

Table 4: Examples of the pairwise alignment and unary salience learned by ALIGNER. Three most salient query tokens and their top-1 pairwise-aligned document tokens are indicated with the same color. We highlight top 50% query tokens and 20% document tokens according to their salience.

## 4.4 INTERPRETABILITY

Table 4 shows examples of the pairwise alignment and unary salience learned by ALIGNER. The model aligns query tokens to contextually similar document tokens, but not necessarily identical tokens. The salience features are also highlighted where important noun phrases and verbs have higher salience, consistent with human intuition. We show more examples of alignments in the Appendix A.3. In general, we observe question answering tasks usually require fewer alignments, while other tasks that require a broad understanding of the document favor larger number of alignments.

## 5 RELATED WORK

Recent research on information retrieval often improves the retrieval accuracy with contrastive pre-training (Ni et al., 2021; Izacard et al., 2022; Oguz et al., 2022), model-based hard negative mining (Xiong et al., 2020; Lu et al., 2021; Qu et al., 2021) and knowledge distillation (Santhanam et al., 2021; Zhang et al., 2022; Reddi et al., 2021). Retrieval efficiency is improved via quantization (Santhanam et al., 2021) or lower-dimensional vectors (Hofstätter et al., 2022).

Term importance and salience have a long history in information retrieval: from term frequency (*tf*) and inverse document frequency (*idf*), to recent BERT-based importance measures (Dai & Callan, 2020; Zhao et al., 2021; Formal et al., 2021b;a). These works mostly focus on sparse lexical retrieval and learn term weights for sparse bag-of-words representations. Our work is most related to ColBERTer (Hofstätter et al., 2022), which proposed to fuse single-vector retrieval and multi-vector refinement. While ColBERTer prunes multi-vector word embeddings for refinement and tests on in-domain retrieval task, we propose to prune multi-vector embeddings for retrieval and mainly study the generalization of retrieval on out-of-domain retrieval tasks. Zhou & Devlin (2021) proposed a multi-vector attention model for reranking, we have a similar formulation but focus on the retrieval.

Recently, Promptagator (Dai et al., 2022b) points out the importance of using a few annotated examples to adapt to a new retrieval task. Promptagator achieves few-shot task adaptation via query generation (Ma et al., 2021; Lee et al., 2021b; Dai et al., 2022a) using large language models (Sanh et al., 2022; Brown et al., 2020; Wei et al., 2022), which has high inference cost. ALIGNER is more versatile and can be fast adapted to a new task via few-shot alignment adaptation.

## 6 CONCLUSION

In this paper, we introduce ALIGNER, a novel sparse alignment method for multi-vector document retrieval. We first formulate different retrieval models with token-level sparse alignments and propose ALIGNER to tackle the limitations of existing models. Specifically, ALIGNER uses pairwise alignments and unary saliences that allow us to adapt to different tasks and prune unimportant tokens, respectively. As a result, we achieve strong performance on both zero-shot and few-shot document retrieval tasks while drastically improving the run-time and storage complexity of multi-vector retrieval. With its interpretable alignments and better performance with large language models, we envision that our multi-vector retrieval model can serve as a strong standalone retriever in the future.

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

# A APPENDIX

## A.1 DERIVATION OF THE ITERATIVE UPDATES

We present the derivation of Eq.5 for solving optimization problem (4) in Section 3.2. The maximization problem (4) can be written as an equivalent minimization problem:

$$
\begin{aligned}
&\max_{\boldsymbol{\lambda}} \quad \boldsymbol{s}^{\top}\boldsymbol{\lambda} + \varepsilon H(\boldsymbol{\lambda}) \\
\iff \quad &\min_{\boldsymbol{\lambda}} \quad -\boldsymbol{s}^{\top}\boldsymbol{\lambda} - \varepsilon H(\boldsymbol{\lambda}) \\
\iff \quad &\min_{\boldsymbol{\lambda}} \quad -\boldsymbol{s}^{\top}\boldsymbol{\lambda} - \varepsilon H(\boldsymbol{\lambda}) - \varepsilon \mathbf{1}^{\top}\boldsymbol{\lambda} \quad\quad (6)\\
&\text{s.t. } \mathbf{1}^{\top}\boldsymbol{\lambda} = k, \ \lambda_i \in [0,1], \ i = 1, \dots, m.
\end{aligned}
$$

Note the term $\varepsilon \mathbf{1}^{\top}\boldsymbol{\lambda}$ will be a constant $\varepsilon \times k$, but we include it in the minimization object to make our derivation simpler later.

Now, let $a \in \mathbb{R}$ and $\boldsymbol{b} \in \mathbb{R}^m$ be the Lagrangian variables corresponding to the linear constraints $\mathbf{1}^{\top}\boldsymbol{\lambda} = k$ and $\lambda_i \leq 1 \ \forall i$ .[8] The minimization problem is equivalent to its Lagrangian expression:

$$
\min_{\boldsymbol{\lambda}\in\mathbb{R}^m} \max_{a\in\mathbb{R}, \boldsymbol{b}\leq\mathbf{0}} -\boldsymbol{s}^{\top}\boldsymbol{\lambda} - \varepsilon H(\boldsymbol{\lambda}) - \varepsilon \mathbf{1}^{\top}\boldsymbol{\lambda} + a(k - \mathbf{1}^{\top}\boldsymbol{\lambda}) + \boldsymbol{b}^{\top}(\mathbf{1} - \boldsymbol{\lambda}) \quad\quad (7)
$$

The objective function (6) is strongly convex and the solution space of $\boldsymbol{\lambda}$ is a convex set. As a result, strong duality holds and we can instead solve the dual problem that exchanges the min and max operators in (7)

$$
\max_{a\in\mathbb{R}, \boldsymbol{b}\leq\mathbf{0}} \min_{\boldsymbol{\lambda}\in\mathbb{R}^m} -\boldsymbol{s}^{\top}\boldsymbol{\lambda} - \varepsilon H(\boldsymbol{\lambda}) - \varepsilon \mathbf{1}^{\top}\boldsymbol{\lambda} + a(k - \mathbf{1}^{\top}\boldsymbol{\lambda}) + \boldsymbol{b}^{\top}(\mathbf{1} - \boldsymbol{\lambda}) \quad\quad (8)
$$

The optimal solution $(a, \boldsymbol{b}, \boldsymbol{\lambda})$ must have the Karush-Kuhn-Tucker (KKT) conditions hold (Kuhn & Tucker, 2014), namely

$$
\frac{\partial\left(-\boldsymbol{s}^{\top}\boldsymbol{\lambda} - \varepsilon H(\boldsymbol{\lambda}) + \varepsilon \mathbf{1}^{\top}\boldsymbol{\lambda} + a(k - \mathbf{1}^{\top}\boldsymbol{\lambda}) + \boldsymbol{b}^{\top}(\mathbf{1} - \boldsymbol{\lambda})\right)}{\partial\boldsymbol{\lambda}} = 0
$$

$$
\iff \quad \boldsymbol{\lambda} = \exp\left(\frac{\boldsymbol{s} + a + \boldsymbol{b}}{\varepsilon}\right) \quad \iff \quad \lambda_i = \exp\left(\frac{s_i + a + b_i}{\varepsilon}\right) \ \ \forall i = 1, \dots, m
$$

Substituting $\boldsymbol{\lambda}$ using the above equation in (8), the dual problem now has a simple form:

$$
\max_{a\in\mathbb{R}, \boldsymbol{b}\leq\mathbf{0}} k \cdot a + \mathbf{1}^{\top}\boldsymbol{b} - \mathbf{1}^{\top}\exp\left(\frac{\boldsymbol{s} + a + \boldsymbol{b}}{\varepsilon}\right)
$$

We can solve this problem using coordinate descent (Wright, 2015) by successively maximizing the function with either $a$ or $\boldsymbol{b}$ fixed. This leads to the iterative updates (Eq.5) described in Section 3.2.

$$
\begin{aligned}
a' &= \varepsilon \ln(k) - \varepsilon \ln\left\{\sum_i \exp\left(\frac{s_i + b_i}{\varepsilon}\right)\right\} \\
b_i' &= \min(-s_i - a', 0)
\end{aligned}
$$

**Discussion** In short, we solve the dual problem of optimization (4) by performing coordinate descent of the dual variables $a$ and $\boldsymbol{b}$. That is, we find the optimal $a$ that maximizes the dual objective given a fixed $\boldsymbol{b}$, and vice versa.

This iterative algorithm is also closely related to the Sinkhorn algorithm of Optimal Transport (OT). In fact, Sinkhorn algorithm solves the entropy-regularized version of Optimal Transport (Cuturi, 2013). However, our work concerns an different optimization instance. While OT solves a transportation problem where the solution space is defined with the marginal constraints over the rows and columns of a transportation matrix, our optimization problem is constrained with a total budget ($\sum_i \lambda_i = k$) and upper bounds ($\lambda_i \leq 1 \ \forall i$). This leads to different iterative updates.

---

[8] $\lambda_i \geq 0 \ \forall i$ is already implied by the entropy term $H(\boldsymbol{\lambda})$ in the objective.

## A.2 DIFFERENTIABLE ALIGNMENT WITH SPARSITY CONSTRAINTS

Besides the Top-$k$ and Top-$p$ alignments in §3.1, we also explore a differentiable pairwise alignment with sparsity contraints (DA). Both Top-$k$ adn Top-$p$ are doing hard selection of alignments, i.e., $\tilde{A}_{i,j}$ is either 1 or 0. We relax it by introducing soft sparsity constraints. Similar to our formulation for unary salience (§3.2), we determine the alignment $\tilde{A}$ by the following optimization problem:

$$
\begin{aligned}
\max_{A} \ & \langle S, A \rangle + \varepsilon H(A) \\
\text{s.t.} \ & \sum_j A_{i,j} = k, \ i = 1, \ldots, n \\
& A_{i,j} \in [0,1], \ i = 1, \ldots, n, \ j = 1, \ldots, m
\end{aligned}
\tag{9}
$$

where $H(\cdot)$ is the elementwise entropy function and $\varepsilon > 0$ is a small constant. We constrain the sum of each row of $\tilde{A}$ to equal $k$. When $\varepsilon = 0$, the solution of Eq. 9 is the same as Top-$k$. When $\varepsilon > 0$, the entropy term makes the optimization problem strongly concave, which can be solved by the same algorithm in Appendix A.1. The solution is differentiable, thus can be trained end-to-end in our model.

## A.3 QUALITATIVE ANALYSIS

| Dataset | Query | Gold Document |
|---|---|---|
| Quora | what is the best birthday gift for a friend? | what[3] is[2] a[4] good[1] birthday gift for a friend? |
| MS MARCO (dev) | when would you use a fathom measurement | a fathom[1] is a unit of length in the imperial and the u.s. customary systems equal to 6 feet[3] (1.8288 m[4]), used especially for measuring the depth of water. there are two yards (6 feet) in an imperial fathom[2]. |
| Touché-2020 | should animals be used for scientific or commercial testing? | animal testing should not be allowed...*[truncated]*...albeit the non-precocious mistakes of scientists[2]. also...*[truncated]*...skeptic of the scientist[3] in question's abilities ...*[truncated]*...continuous use if animals for clinical[4] and basic research." ...*[truncated]*...majority of the scientific[1] community thinks on this issue, ... |

Table 5: Examples of pairwise alignment with the top-$k$ value up to 4 for the Quora, MS MARCO, and Touché-2020 datasets. Query tokens being aligned are shown in blue, and corresponding aligned document tokens are shown in red. The superscript on the document token $(k)$ indicates top-$k$ alignment. We notice that the top-1 alignment quality is generally good across all three tasks. However, larger value of $k$ results in spurious irrelevant alignments for Quora and MS MARCO, while remains fairly useful for Touché-2020.

Table 5 shows examples of top-$k$ pairwise alignments of a query token (highlighted in blue) to the corresponding document tokens for several different tasks. For question-answering (e.g., MS MARCO) and duplicate question retrieval (Quora), fewer alignments seem to be preferable, and as $k$ increases, we start to see spurious alignments to unrelated documents tokens. For argument retrieval tasks such as Touché-2020, on the other hand, larger value of $k$ tends to provide useful semantically relevant alignments (e.g., *scientific* vs *clinical*). These qualitative examples provide intuitive insights regarding why different alignment strategies are helpful for different tasks, and why alignment adaptation is necessary.

## A.4 RESULTS ON MS MARCO

Table 6 shows the retrieval performance of ALIGNER and previous models on the MS MARCO dev set. We deliberately kept the training configuration of ALIGNER relatively simpler (e.g., no distillation or model-based hard negatives). However, ALIGNER still achieves the best MRR@10 simply because of scaling to larger pretrained language models. We have also trained ALIGNER with other alignment strategies, such as top-4, top-1%, and DA (Appendix A.2). However, the results suggest top-1 is favorable in MS MARCO.

| Model | MRR@10 | Recall@1000 |
|---|---|---|
| BM25 | 18.7 | 85.7 |
| SPLADE$_{v2}$ | 36.8 | 97.9 |
| DPR | 31.1 | 95.2 |
| GTR$_{base}$ | 36.6 | 98.3 |
| GTR$_{large}$ | 37.9 | **99.1** |
| GTR$_{xl}$ | 38.5 | 98.9 |
| GTR$_{xxl}$ | 38.8 | 99.0 |
| ColBERT | 36.0 | 96.8 |
| ColBERT$_{v2}$ | 39.7 | 98.4 |
| COIL | 35.5 | 96.3 |
| ME-BERT | 33.4 | – |
| ALIGNER$_{base}$ | 38.8 | 97.8 |
| ALIGNER$_{large}$ | 39.4 | 98.3 |
| ALIGNER$_{xl}$ | 39.9 | 98.4 |
| ALIGNER$_{xxl}$ | **40.3** | 98.7 |
| ALIGNER$_{base}$ (top-4) | 37.1 | 97.5 |
| ALIGNER$_{base}$ (top-1%) | 38.8 | 97.6 |
| ALIGNER$_{base}$ (DA) | 37.8 | 97.4 |

Table 6: Retrieval performance on MS MARCO. The top half shows baselines from previous work. The botton half shows different ALIGNER models. DA: differential alignment. See Appendix A.2

## A.5 FULL RESULT TABLES ON BEIR

Tables 7 to 10 presents complete results of ALIGNER's performance on the BEIR datasets initialized from T5 base, large, XL, and XXL checkpoints. We set $k = 1$ during training, and show results across different inference-time alignment strategies (both top-$k$ and top-$p$). As expected, model accuracy improves as we scale to larger models. Moreover, we observe similar benefits of alignment adaptation across all the different model sizes.

| ALIGNER$_{base}$ | | Top-$k$ | | | | | Top-$p$ | | | |
|---|---|---|---|---|---|---|---|---|---|---|
| | | 1* | 2 | 4 | 6 | 8 | 0.5% | 1% | 1.5% | 2% |
| A.R. | ArguAna | 28.8 | 24.4 | 18.3 | 14.5 | 11.4 | 33.3 | 45.5 | 48.1 | 46.9 |
| | Touché-2020 | 34.8 | 50.0 | 51.1 | 49.3 | 46.0 | 31.3 | 24.2 | 20.3 | 15.8 |
| F.C. | FEVER | 72.4 | 75.0 | 68.3 | 57.6 | 49.0 | 72.5 | 55.0 | 44.9 | 29.9 |
| | Climate-FEVER | 18.1 | 20.8 | 23.0 | 23.0 | 22.7 | 18.2 | 13.7 | 13.8 | 10.8 |
| | SciFact | 70.4 | 68.8 | 65.0 | 60.9 | 55.6 | 71.1 | 69.4 | 67.2 | 62.7 |
| Q.A. | NQ | 52.2 | 48.3 | 36.3 | 26.6 | 19.9 | 52.2 | 49.2 | 43.5 | 36.3 |
| | HotpotQA | 61.7 | 58.6 | 36.0 | 21.9 | 13.9 | 61.7 | 60.1 | 54.3 | 47.3 |
| | FiQA | 33.4 | 30.8 | 23.8 | 19.5 | 15.5 | 33.4 | 28.5 | 24.6 | 19.0 |
| | BioASQ | 49.6 | 45.8 | 37.4 | 30.8 | 24.5 | 47.4 | 37.7 | 31.9 | 25.0 |
| | NFCorpus | 34.0 | 33.2 | 32.0 | 29.9 | 27.8 | 33.6 | 31.7 | 30.5 | 28.6 |
| MISC. | TREC-COVID | 68.3 | 74.0 | 73.2 | 67.2 | 61.7 | 66.8 | 64.4 | 56.7 | 46.7 |
| | SCIDOCS | 14.1 | 14.3 | 13.1 | 11.4 | 9.7 | 14.4 | 14.9 | 14.9 | 14.8 |
| | DBPedia | 41.6 | 39.4 | 29.6 | 20.2 | 14.2 | 41.6 | 41.7 | 39.9 | 36.6 |
| | Quora | 82.3 | 64.9 | 30.6 | 13.3 | 6.3 | 82.3 | 82.3 | 82.3 | 82.3 |
| | Average | 47.3 | 46.3 | 38.4 | 31.9 | 27.0 | 47.1 | 44.2 | 40.9 | 35.9 |

Table 7: nDCG@10 on the BEIR benchmark with different $k$ and $p$ in ALIGNER$_{base}$. *: alignment strategy during training ($k = 1$).

| ALIGNER$_{large}$ | | Top-$k$ | | | | | Top-$p$ | | | |
|---|---|---|---|---|---|---|---|---|---|---|
| | | 1* | 2 | 4 | 6 | 8 | 0.5% | 1% | 1.5% | 2% |
| A.R. | ArguAna | 29.5 | 25.2 | 19.5 | 16.0 | 13.6 | 33.2 | 44.5 | 47.9 | 46.8 |
| | Touché-2020 | 36.7 | 47.5 | 53.0 | 53.4 | 52.0 | 32.5 | 25.4 | 20.7 | 16.3 |
| F.C. | FEVER | 72.9 | 75.2 | 69.6 | 61.8 | 53.6 | 72.8 | 52.8 | 43.0 | 29.3 |
| | Climate-FEVER | 18.6 | 20.7 | 23.3 | 23.4 | 23.5 | 18.6 | 13.1 | 14.2 | 11.4 |
| | SciFact | 71.5 | 70.7 | 69.5 | 67.7 | 64.1 | 72.2 | 70.6 | 69.6 | 66.8 |
| Q.A. | NQ | 57.2 | 52.8 | 43.4 | 36.3 | 31.2 | 57.2 | 53.7 | 47.5 | 41.0 |
| | HotpotQA | 63.6 | 62.6 | 44.8 | 32.1 | 24.4 | 63.6 | 61.8 | 55.6 | 48.5 |
| | FiQA | 39.4 | 35.6 | 30.4 | 26.2 | 23.4 | 39.4 | 33.3 | 28.4 | 22.8 |
| | BioASQ | 53.3 | 50.7 | 43.1 | 36.8 | 31.6 | 49.2 | 38.2 | 33.2 | 27.4 |
| | NFCorpus | 35.5 | 33.9 | 32.5 | 31.2 | 29.6 | 34.6 | 32.1 | 30.8 | 29.6 |
| MISC. | TREC-COVID | 71.9 | 79.4 | 77.3 | 74.4 | 69.3 | 70.0 | 66.1 | 59.7 | 52.4 |
| | SCIDOCS | 15.3 | 15.5 | 14.9 | 13.6 | 12.5 | 15.4 | 15.8 | 16.0 | 15.8 |
| | DBPedia | 43.5 | 41.9 | 34.7 | 29.0 | 24.3 | 43.5 | 43.5 | 41.5 | 37.4 |
| | Quora | 84.5 | 75.5 | 46.4 | 20.9 | 8.4 | 84.5 | 84.5 | 84.5 | 84.5 |
| | Average | 49.5 | 49.1 | 43.0 | 37.3 | 33.0 | 49.0 | 45.4 | 42.3 | 37.9 |

Table 8: nDCG@10 on the BEIR benchmark with different $k$ and $p$ in ALIGNER$_{large}$. *: alignment strategy during training ($k = 1$).

| ALIGNER$_{xl}$ | | Top-$k$ | | | | | Top-$p$ | | | |
|---|---|---|---|---|---|---|---|---|---|---|
| | | 1* | 2 | 4 | 6 | 8 | 0.5% | 1% | 1.5% | 2% |
| A.R. | ArguAna | 32.4 | 28.3 | 22.5 | 18.9 | 16.3 | 35.3 | 44.9 | 47.2 | 47.0 |
| | Touché-2020 | 36.2 | 46.4 | 53.2 | 51.0 | 50.5 | 33.0 | 26.1 | 21.9 | 18.2 |
| F.C. | FEVER | 72.9 | 75.6 | 70.9 | 63.2 | 56.0 | 72.9 | 56.9 | 48.2 | 34.1 |
| | Climate-FEVER | 18.7 | 21.2 | 23.2 | 23.9 | 24.2 | 18.8 | 15.2 | 16.6 | 14.3 |
| | SciFact | 71.5 | 70.8 | 69.7 | 67.6 | 65.6 | 73.0 | 71.6 | 69.9 | 69.5 |
| Q.A. | NQ | 58.8 | 54.9 | 46.0 | 39.5 | 34.1 | 58.8 | 55.5 | 50.1 | 43.8 |
| | HotpotQA | 63.9 | 62.6 | 45.5 | 32.9 | 25.0 | 63.9 | 62.5 | 57.5 | 51.4 |
| | FiQA | 40.8 | 37.4 | 32.2 | 28.9 | 25.2 | 40.8 | 35.0 | 31.2 | 25.7 |
| | BioASQ | 53.6 | 50.4 | 43.0 | 36.6 | 32.2 | 50.2 | 39.7 | 34.4 | 28.9 |
| | NFCorpus | 35.4 | 34.3 | 33.0 | 31.5 | 29.7 | 35.0 | 33.1 | 31.6 | 30.0 |
| MISC. | TREC-COVID | 75.1 | 80.6 | 80.5 | 78.1 | 73.5 | 72.5 | 69.4 | 62.0 | 54.0 |
| | SCIDOCS | 15.4 | 16.0 | 15.5 | 14.3 | 13.3 | 15.6 | 16.2 | 16.5 | 16.4 |
| | DBPedia | 43.6 | 42.2 | 36.2 | 30.6 | 26.5 | 43.6 | 43.6 | 42.4 | 39.4 |
| | Quora | 85.3 | 76.5 | 50.1 | 26.9 | 13.1 | 85.3 | 85.3 | 85.3 | 85.3 |
| | Average | 50.3 | 49.8 | 44.4 | 38.9 | 34.7 | 49.9 | 46.8 | 43.9 | 39.8 |

Table 9: nDCG@10 on the BEIR benchmark with different $k$ and $p$ in ALIGNER$_{xl}$. *: alignment strategy during training ($k = 1$).

| ALIGNER$_{xxl}$ | | Top-$k$ | | | | | Top-$p$ | | | |
|---|---|---|---|---|---|---|---|---|---|---|
| | | 1* | 2 | 4 | 6 | 8 | 0.5% | 1% | 1.5% | 2% |
| A.R. | ArguAna | 33.8 | 29.6 | 24.1 | 20.4 | 18.2 | 36.6 | 46.9 | 49.8 | 49.7 |
| | Touché-2020 | 34.5 | 47.4 | 49.8 | 51.1 | 50.6 | 30.1 | 21.8 | 16.3 | 11.7 |
| F.C. | FEVER | 74.2 | 77.0 | 73.9 | 67.9 | 62.2 | 75.2 | 47.1 | 36.2 | 24.1 |
| | Climate-FEVER | 19.7 | 21.6 | 23.7 | 23.2 | 24.3 | 19.7 | 12.0 | 12.3 | 9.3 |
| | SciFact | 73.1 | 71.2 | 71.3 | 69.1 | 67.0 | 74.4 | 71.5 | 69.9 | 69.5 |
| Q.A. | NQ | 60.5 | 56.0 | 49.1 | 44.0 | 40.1 | 60.4 | 54.5 | 45.8 | 37.9 |
| | HotpotQA | 65.2 | 63.4 | 48.8 | 37.7 | 30.1 | 65.2 | 63.0 | 56.0 | 48.1 |
| | FiQA | 43.5 | 40.3 | 36.8 | 33.8 | 31.4 | 43.5 | 35.9 | 30.4 | 24.0 |
| | BioASQ | 54.8 | 51.1 | 43.3 | 38.2 | 34.4 | 49.6 | 35.8 | 29.4 | 24.9 |
| | NFCorpus | 35.2 | 34.0 | 32.6 | 31.5 | 30.3 | 34.1 | 29.7 | 28.0 | 27.1 |
| MISC. | TREC-COVID | 75.8 | 81.4 | 80.1 | 75.6 | 71.9 | 75.1 | 65.4 | 54.0 | 45.9 |
| | SCIDOCS | 17.1 | 16.8 | 16.3 | 15.6 | 14.2 | 17.1 | 17.0 | 16.6 | 15.9 |
| | DBPedia | 45.0 | 43.2 | 38.3 | 33.9 | 30.6 | 45.0 | 44.9 | 42.6 | 35.7 |
| | Quora | 86.0 | 79.0 | 58.6 | 38.2 | 22.8 | 86.0 | 86.0 | 86.0 | 86.0 |
| | Average | 51.3 | 50.9 | 46.2 | 41.4 | 37.7 | 50.8 | 45.1 | 40.9 | 36.4 |

Table 10: nDCG@10 on the BEIR benchmark with different $k$ and $p$ in ALIGNER$_{xxl}$. *: alignment strategy during training ($k = 1$).

| ALIGNER$_{base}$ | | Top-$k$ | | | | | Top-$p$ | | | |
|---|---|---|---|---|---|---|---|---|---|---|
| | | 1 | 2 | 4* | 6 | 8 | 0.5% | 1% | 1.5% | 2% |
| A.R. | ArguAna | 26.5 | 23.6 | 18.8 | 15.9 | 13.6 | 30.7 | 42.8 | 45.7 | 46.2 |
| | Touché-2020 | 25.2 | 32.4 | 43.9 | 48.2 | 49.5 | 19.7 | 14.6 | 11.9 | 9.5 |
| F.C. | FEVER | 59.0 | 67.0 | 72.8 | 73.6 | 73.0 | 59.0 | 34.3 | 25.4 | 18.4 |
| | Climate-FEVER | 14.7 | 17.5 | 20.9 | 22.8 | 23.7 | 14.7 | 7.8 | 7.1 | 5.1 |
| | SciFact | 70.1 | 70.6 | 70.4 | 69.2 | 68.0 | 69.6 | 66.7 | 65.6 | 64.9 |
| Q.A. | NQ | 42.3 | 49.4 | 52.4 | 51.2 | 49.1 | 42.3 | 37.3 | 30.9 | 25.3 |
| | HotpotQA | 57.6 | 61.0 | 60.4 | 57.4 | 53.0 | 57.6 | 55.3 | 49.4 | 43.4 |
| | FiQA | 29.8 | 32.2 | 32.0 | 30.1 | 27.8 | 29.9 | 22.2 | 17.7 | 14.0 |
| | BioASQ | 48.3 | 50.9 | 49.0 | 48.0 | 45.9 | 41.1 | 26.4 | 21.1 | 17.4 |
| | NFCorpus | 31.8 | 33.5 | 33.9 | 33.6 | 33.0 | 29.9 | 26.3 | 25.7 | 25.2 |
| MISC. | TREC-COVID | 54.8 | 63.5 | 72.9 | 75.4 | 74.5 | 53.1 | 48.7 | 42.5 | 38.1 |
| | SCIDOCS | 13.2 | 14.1 | 14.7 | 14.6 | 14.0 | 13.2 | 12.2 | 11.7 | 11.5 |
| | DBPedia | 31.5 | 39.1 | 42.1 | 39.9 | 36.9 | 31.4 | 31.4 | 29.7 | 25.1 |
| | Quora | 82.7 | 80.5 | 72.1 | 57.4 | 41.9 | 82.7 | 82.7 | 82.7 | 82.7 |
| | Average | 42.0 | 45.4 | 46.9 | 45.5 | 43.1 | 41.1 | 36.3 | 33.4 | 30.5 |

Table 11: nDCG@10 on the BEIR benchmark with different $k$ and $p$ in ALIGNER$_{base}$. *: alignment strategy during training ($k = 4$).

| ALIGNER$_{base}$ | | Top-$k$ | | | | | Top-$p$ | | | |
|---|---|---|---|---|---|---|---|---|---|---|
| | | 1 | 2 | 4 | 6 | 8 | 0.5% | 1%* | 1.5% | 2% |
| A.R. | ArguAna | 28.4 | 23.7 | 17.9 | 14.2 | 11.1 | 32.9 | 45.3 | 48.0 | 47.7 |
| | Touché-2020 | 38.1 | 51.3 | 53.0 | 51.3 | 50.0 | 35.0 | 26.6 | 22.6 | 17.1 |
| F.C. | FEVER | 72.1 | 74.6 | 67.7 | 57.5 | 49.2 | 72.2 | 56.1 | 46.2 | 31.2 |
| | Climate-FEVER | 17.8 | 20.3 | 22.0 | 22.1 | 21.8 | 17.8 | 13.5 | 14.4 | 11.3 |
| | SciFact | 69.0 | 67.9 | 64.8 | 61.5 | 57.2 | 70.5 | 70.3 | 68.1 | 63.9 |
| Q.A. | NQ | 52.6 | 48.2 | 36.4 | 26.4 | 20.2 | 52.5 | 50.4 | 44.7 | 38.2 |
| | HotpotQA | 60.6 | 57.7 | 35.1 | 21.1 | 13.1 | 60.7 | 59.6 | 53.9 | 47.6 |
| | FiQA | 33.7 | 29.4 | 23.5 | 18.8 | 15.8 | 33.6 | 29.8 | 26.2 | 20.6 |
| | BioASQ | 40.3 | 48.4 | 39.9 | 33.1 | 26.4 | 50.0 | 40.3 | 35.3 | 28.8 |
| | NFCorpus | 34.7 | 33.7 | 32.1 | 30.2 | 27.8 | 34.3 | 32.7 | 31.6 | 29.6 |
| MISC. | TREC-COVID | 67.6 | 73.4 | 73.1 | 67.7 | 62.5 | 67.2 | 65.7 | 60.6 | 52.0 |
| | SCIDOCS | 13.9 | 14.2 | 13.1 | 11.5 | 9.9 | 14.1 | 14.6 | 14.8 | 14.9 |
| | DBPedia | 41.3 | 40.3 | 29.5 | 21.2 | 14.0 | 41.2 | 41.3 | 40.3 | 37.1 |
| | Quora | 83.6 | 60.8 | 22.8 | 8.9 | 4.3 | 83.6 | 83.6 | 83.6 | 83.6 |
| | Average | 43.6 | 42.9 | 35.4 | 29.7 | 25.5 | 44.4 | 42.0 | 39.3 | 34.9 |

Table 12: nDCG@10 on the BEIR benchmark with different $k$ and $p$ in ALIGNER$_{base}$. *: alignment strategy during training ($p = 1\%$).

| ALIGNER$_{base}$ | | $k$ | | | | | $\varepsilon$ | | | | |
|---|---|---|---|---|---|---|---|---|---|---|---|
| | | 1* | 2 | 4 | 6 | 8 | 0.01 | 0.02 | 0.04 | 0.06 | 0.1 |
| A.R. | ArguAna | 30.8 | 26.3 | 20.1 | 15.8 | 12.4 | 30.8 | 33.2 | 38.8 | 43.5 | 50.1 |
| | Touché-2020 | 37.4 | 50.1 | 52.0 | 51.1 | 49.0 | 37.4 | 35.0 | 29.8 | 26.9 | 19.4 |
| F.C. | FEVER | 68.8 | 71.4 | 63.7 | 46.1 | 44.3 | 68.8 | 68.0 | 66.4 | 64.5 | 56.2 |
| | Climate-FEVER | 16.7 | 19.4 | 21.2 | 21.6 | 21.3 | 16.7 | 16.2 | 15.6 | 15.6 | 15.4 |
| | SciFact | 69.8 | 68.2 | 64.7 | 61.8 | 57.0 | 69.8 | 69.5 | 70.3 | 71.0 | 70.1 |
| Q.A. | NQ | 52.0 | 47.4 | 35.0 | 25.8 | 18.5 | 52.0 | 51.6 | 50.8 | 50.0 | 46.0 |
| | HotpotQA | 59.6 | 56.1 | 33.4 | 20.6 | 12.6 | 59.6 | 59.5 | 60.0 | 61.0 | 59.8 |
| | FiQA | 32.9 | 29.6 | 23.0 | 18.6 | 14.5 | 32.9 | 32.7 | 32.9 | 32.4 | 30.9 |
| | BioASQ | 50.4 | 47.2 | 38.0 | 31.4 | 24.5 | 50.4 | 50.4 | 50.8 | 50.4 | 43.7 |
| | NFCorpus | 34.1 | 33.6 | 31.6 | 29.6 | 27.4 | 34.1 | 34.4 | 34.2 | 34.2 | 33.5 |
| MISC. | TREC-COVID | 63.9 | 73.2 | 72.5 | 67.7 | 62.8 | 63.9 | 65.5 | 63.6 | 63.8 | 54.2 |
| | SCIDOCS | 14.0 | 14.4 | 13.4 | 11.6 | 9.6 | 14.0 | 14.1 | 14.2 | 14.5 | 15.4 |
| | DBPedia | 40.9 | 38.7 | 28.7 | 19.9 | 13.2 | 40.9 | 40.5 | 39.9 | 39.7 | 38.6 |
| | Quora | 82.8 | 61.2 | 21.5 | 7.4 | 3.5 | 82.8 | 83.1 | 83.5 | 84.0 | 85.0 |
| | Average | 43.6 | 42.4 | 34.6 | 28.6 | 24.7 | 43.6 | 43.6 | 43.4 | 43.4 | 41.2 |

Table 13: nDCG@10 on the BEIR benchmark with ALIGNER$_{base}$ and differentiable alignment (Appendix A.2). *: alignment strategy during training ($k = 1$).

