# OpenReview forum: "Multi-Vector Retrieval as Sparse Alignment"
_ICLR.cc/2023/Conference — Submitted to ICLR 2023_

### Official Review · Reviewer_5rDb · 2022-10-24

**Confidence:** 4
**Correctness:** 4
**Technical Novelty And Significance:** 3
**Empirical Novelty And Significance:** 3
**Recommendation:** 6

**Clarity, Quality, Novelty And Reproducibility:**

The paper is clear and of good shape with reasonable reproducibility. It will be good if the author publish training and test code.

**Strength And Weaknesses:**

Strengths: (1) novel perspective on unifying previous multi-vector document retrieval and build the sparsified end-to-end scheme on top of them.
(2) Good efficiency trade off on the sparsity side
(3) Good results on the  BEIR benchmark


Weakness: (1) entropy-regularized linear programming? The paper does not explain why  entropy-regularization on the linear programming is a desired choice. For enabling end-to-end training, a quadratic term would also work?
(2) The paper does not explain why end-to-end training in the  entropy-regularization is necessary. During training, isn't the  model uses top-1 alignment where the only one document token is activated for each query token.
(3) As the sparsity of the training and test can be different, it is good to investigating optimal training sparsity.
(4) When training with top-1 alignment, the model looks similar to ColBERT except for the unary masking. It is good to analyze how and why the  unary masking is beneficial

**Summary Of The Paper:**

The paper leads a discussion on multi-vector document retrieval by introducing a sparse alignment perspective that can include serval previous document retrieval scheme. On top of that, the paper proposes an entropy-regularized sparse relaxation of the novel alignment scheme by separating the query-document token-wise similarity scores to a similarity times unary encoding of the query and the document . The advantage of that is to improve efficiency, avoiding costly computation on n*m matrix of the similarity. It is worth mentioning that the sparsity controls can be different between training and test. During training, the paper uses top-1 and during test, the paper find optimal sparsity using cross validation.

**Summary Of The Review:**

The paper provides a novel perspective on unifying various multi-vector document retrieval scheme under a same framework and proposed a sparse alignment on top of the framework. The idea is interesting and novel however is still with some under-explained concerns of mine

---

> ### Author Response · Authors · 2022-11-19
> **Response to Reviewer 5rDb (Part 1)**
>
> Thanks to the reviewer for finding our work novel and our method effective. We address the reviewer's concerns in the following order.
>
> > The paper does not explain why end-to-end training in the entropy-regularization is necessary. During training, isn't the model uses top-1 alignment where the only one document token is activated for each query token.
>
> We would like to clarify that entropy regularized linear programming is used for unary salience prediction rather than pairwise alignment and top-1 alignment is used in the final model. In fact, we have an extensive study on how the design of pairwise alignment affects retrieval performance, including using entropy regularized linear programming for pairwise alignment (Appendix A.2). In the latest manuscript, more models trained with different pairwise alignment are included in Figure 4. It is observed that the models trained with top-k=1, topk=2, and top-p=0.5% are competitive, while models trained with top-k=8 and top-p=4% have the worst performance, hypothetically due to the characteristics of the MS MARCO training data. Moreover, alignment adaptation can close the gap among these models, indicating the effectiveness of the method for few-shot retrieval.
>
> > When training with top-1 alignment, the model looks similar to ColBERT except for the unary masking. It is good to analyze how and why the unary masking is beneficial
>
> The main benefit of unary salience is that it enables AligneR to improve multi-vector retrieval efficiency (Section 3.2). And retrieval efficiency is one of the major blockers preventing multi-vector models from being widely adopted in practice. In Figure 6, we showed that using salience prediction our model can prune up to 70% query tokens and 80% document tokens with minimal performance loss on MS MARCO. This is equivalent to an 80% reduction in multi-vector retrieval index and a 94% reduction in computation (assume brute force search). In Figure 7, it is observed that token pruning by salience can even improve retrieval performance for some out-of-domain retrieval tasks, such as Touche-2020, TREC-covid, etc. It is also observed that token pruning hurts the retrieval performance for DBPedia. Generally speaking, AlgineR’s contextualized token embedding, pairwise alignment and unary salience can all generalize to OOD retrieval tasks.
>
> In Figure 4, we show that AligneR models trained with top-k=2 and top-p=0.5% are as competitive as the one trained with top-k=1.
>
> > entropy-regularized linear programming? The paper does not explain why entropy-regularization on linear programming is a desired choice. For enabling end-to-end training, a quadratic term would also work?
>
> Thanks for the great comment. We have included more discussion in Section 3.2 to better motivate the entropy-regularized linear programming method.
>
> Sparsity is an important inductive bias for learning meaningful salience estimations as we do not have explicit supervision for salience. In this work, we achieve sparsity with an entropy regularized linear program, and the motivation behind is to explicitly control the sparsity ratios. While alternative techniques, such as L1 regularization, can also be used, they cannot guarantee the sparsity ratio and we have to empirically determine a coefficient to balance the losses. A quadratic term in the linear program can also enable end-to-end training. Related methods such as L2 regularized LP can be applied as well, as the reviewer pointed out. We choose the entropy-regularized version because it gives very simple closed-form updates shown in Eq.(5), whereas L2 regularized version may require more sophisticated (and custom) forward and backward implementation (see for example https://arxiv.org/pdf/2001.04437.pdf). We also discussed the connection between our method and the entropy-regularized Optimal Transport in Appendix A.
>
> We argue that our main contribution is not this entropy regularized linear program technique, but the idea of sparsifying the salience features and pruning the tokens for indexing and retrieval. Previously, multi-vector retrieval models built an index for all document token vectors, and used each query token vector to do a nearest neighbor search in the index. With salience features, we can prune the tokens and improve the efficiency of the system, and sparsity is the key for the learned salience to be meaningful. We acknowledge that other sparsity methods, such as L1 loss, can also be applied, but it does not mitigate with our contribution.

---

> > ### Author Response · Authors · 2022-11-19
> > **Response to Reviewer 5rDb (Part 2)**
> >
> > > As the sparsity of the training and test can be different, it is good to investigating optimal training sparsity.
> >
> > We thank the reviewer for the suggestion of investigating the optimal training sparsity. We do have the analysis in Figure 4, and further expand it in the revision. Specifically, we explored training with alignment sparsity from top-1 to top-8, and top-0.5% to top-4%. The results suggest that training with top-k=1, top-k=2 and top-p=0.5% alignments works the best for BEIR benchmark.

---

> > ### Comment · Reviewer_5rDb · 2022-11-23
> > **Responses**
> >
> > The author's response address most of my concerns and I would like to raise my score to 6 as marginally above the acceptance threshold

---

> > > ### Author Response · Authors · 2022-11-25
> > > **Thank you for the feedback**
> > >
> > > I am glad that our responses addressed your concerns. Thank you.

---

> ### Author Response · Authors · 2022-11-23
> **Looking forward to your feedback**
>
> Dear reviewer,
>
> We hope that you've had a chance to read our responses. We would really appreciate a reply as to whether our responses and clarifications have addressed the issues raised in the review, or whether there is anything else we can address.

---

### Official Review · Reviewer_4BA2 · 2022-10-25

**Confidence:** 5
**Correctness:** 4
**Technical Novelty And Significance:** 3
**Empirical Novelty And Significance:** 4
**Recommendation:** 6

**Clarity, Quality, Novelty And Reproducibility:**

Clarity: This paper is well-written and the motivation is very clear.
Quality: The quality of the article is good.
Novelty: In general, compared with the previous multi-vector retrieval model, this paper introduces sparse alignment, which is a good attempt.
Reproducibility: It is can be implemented following the description in the paper.


**Strength And Weaknesses:**

Generally speaking, this paper puts forward a universal retrieval scheme, and achieves good results, the specific advantages are as follows:

(1) The sparse alignment of multi-vector retrieval is a good solution to solve the retrieval effect and efficiency.

(2) The new proposed model in this paper has achieved better results than the previous work in many tasks on MSMARCO.

(3) The proposed model in this paper uses sparse alignment matrix to aggregate token-level similarity, where each element represents the alignment of a pair of tokens, which can develop different retrieval models in a unified way and identify the shortcomings of existing models.

However, there is a lack of unified experimental standards and ablation experiments in this paper. So I also have a few doubts about this article:

(1) This paper uses 6B T5-V1.1, but the previous baseline work only GTRxxl has the same size, while ColBERTv2 using multi-vector retrieval model has only 110m model size. However, GTR is a single vector retrieval model, so there is no unified standard to show that the effect of the proposed model in this paper is better than the previous model.

(2) In this paper, similarity and alignment structures are proposed, but no ablation experiments have been carried out to prove the effectiveness of the improved model.

(3) This model still needs to calculate the similarity matrix for queries and documents, so the computational efficiency is not improved compared with other multi-vector retrieval models in the training stage.


**Summary Of The Paper:**

Multi-vector retrieval models can fetch improved results on certain retrieval tasks, while at the same time the approach leads to large search indexes and expensive computation. Thus, this paper proposes ALIGNER, a novel multi-vector retrieval model that learns sparse pairwise alignments between query and document tokens, as well as the unary saliences of each token reflecting their relative importance to the retrieval. The authors show experimentally that controlling the sparsity of pairwise token alignments usually leads to significant performance gains. On the other hand, unary saliences determine whether a token needs to be aligned with other tokens for retrieval. With sparsified unary saliences, it is possible to prune a large number of query and document token vectors and improve the efficiency of multi-vector retrieval. In a zero-sample setup, ALIGNER scores 51.1 points nDCG@10 implements the latest state of the new retrieval-only retriever on 13 tasks in the BEIR benchmark test.

**Summary Of The Review:**

This paper proposes a sparse aligned multi-vector retrieval model, a novel model structure, which achieves good retrieval results and alleviates the shortcomings of previous work to some extent. Despite there is lack of corresponding ablation experiments, I think it is an overall good paper.

---

> ### Author Response · Authors · 2022-11-19
> **Response to Reviewer 4BA2**
>
> Thanks to the review for finding our universal retrieval scheme interesting and our empirical results convincing. We address the reviewer's concerns in the following order.
>
> > This paper uses 6B T5-V1.1, but the previous baseline work only GTRxxl has the same size, while ColBERTv2 using multi-vector retrieval model has only 110m model size. However, GTR is a single vector retrieval model, so there is no unified standard to show that the effect of the proposed model in this paper is better than the previous model.
>
> We thank the reviewer for the comment. Originally, we reported the performance of AligneR_base in Figure 4 and Appendix. We agree with the reviewer that this is not clear and thus we included both AligneR_base and GTR_base in Table 2 for comparison. Both AligneR_base and GTR_base have the same model size as ColBERTv2.
>
> AligneR_base training is the same as GTR_base finetuning (Table 1, Section 4.1), while GTR_base is also pretrained on additional supervision. AligneR_base is significantly better than GTR_base on MS MARCO and BEIR, featuring the superior performance of the multi-vector retriever. AligneR_base performs slightly poorer than ColBERTv2, as we did not use distillation from cross attention models or model-based hard negative mining (section 4.1), which improve model quality but complicate the training and increase computational costs. We adopt a simple training recipe, as our focus is to demonstrate the effect of alignment adaptation and salience pruning. It is worth noting that alignment adaptation consistently improves all AligneR models (Table 2, Figure 4).
>
> > In this paper, similarity and alignment structures are proposed, but no ablation experiments have been carried out to prove the effectiveness of the improved model.
>
> If we understand correctly, the reviewer refers to Figure 1 and Equation 1. In Figure 1, we provided a unified formulation to compare many existing retrieval models. Using the formulation, we would like to clarify that most existing retriever models share the same similarity matrix (there could be small differences such as dot product vs cosine similarity). The major difference among existing methods is the design of the alignment matrix. We propose to study a better design for the alignment matrix A. And our proposed model is illustrated in Figure 2 and Equation 2. In the experiment section 4, we demonstrate the effectiveness of few shot alignment adaptation (Table 2, Table 3, Figure 4, Figure 5). For few-shot retrieval, AligneR outperforms Promptagator which is based on dual encoder. In Figure 6, it is shown that multi-vector retrieval efficiency can be improved by up to 94% thanks to pruning by sparse unary salience. In addition, token pruning is also effective for out-of-domain retrieval (Figure 7).
>
> > This model still needs to calculate the similarity matrix for queries and documents, so the computational efficiency is not improved compared with other multi-vector retrieval models in the training stage.
>
> We acknowledge that in this paper, we mainly concern the multi-vector retrieval efficiency at inference time rather than training efficiency, which is an important factor of an IR system, and is a major drawback of existing multi-vector retrieval models which limits their applicability in the real world. Multi-vector refinement is relatively cheap as token embeddings are cached.
>
> As shown in Table 1, AligneR’s training is in-fact simpler. AligneR is only fine-tuned on MSMARCO + a fixed set of negatives. The fine-tuning is the same as GTR’s fine-tuning while GTR is also pretrained. Another multi-vector model ColBERTv2 is distilled and requires model-based hard negatives. We agree with the reviewer that training efficiency of multi-vector models can be improved, e.g. using a smaller embedding dimension as suggested in ColBERTer. It is a great research direction but not the focus of the presented work.

---

> > ### Comment · Reviewer_4BA2 · 2022-11-28
> > **Thanks**
> >
> > The author's responses address most of my concerns, and I think it is a good paper above the acceptance threshold.

---

> > > ### Author Response · Authors · 2022-11-29
> > > **Thank you for the feedback**
> > >
> > > We are glad that our responses have addressed most of your concerns. We are excited that you agree this is a good paper above the acceptance threshold!

---

> ### Author Response · Authors · 2022-11-23
> **Looking forward to your feedback**
>
> Dear reviewer,
>
> We hope that you've had a chance to read our responses. We would really appreciate a reply as to whether our responses and clarifications have addressed the issues raised in the review, or whether there is anything else we can address.

---

### Official Review · Reviewer_sjAr · 2022-10-25

**Confidence:** 5
**Correctness:** 3
**Technical Novelty And Significance:** 2
**Empirical Novelty And Significance:** 2
**Recommendation:** 6

**Clarity, Quality, Novelty And Reproducibility:**


The paper is mostly clear and well organized. There is some novelty although sparsifying is largely studied in IR.

Section 3.2 (sparsity regularizer) is not well motivated - why not using a L1 loss (like in ColBERTer)? The authors should discuss and motivate much more this aspect since it could interesting for other works (but like it is, the approach is not really convincing). Also, why using two pruning ratios (for training and inference)? How were those ratios determined?

In the experiments,

1) ColBERTer and/or ColBERTv2 should be compared with the approach (or at least, your entropy-based regularization should be compared with a simpler L1 one). Some related works cited in the ColBERTer paper are purely ignored, but quite related to what you are doing (e.g. Multi-Vector Attention Models for Deep Re-ranking, )
2) An even simpler pruning strategy, random pruning, has not been experimented with
3) Some extra-care should be taken to compare comparable models (in terms of parameters at least, a 60x ratio is not acceptable)

Other issues:
- Foonote 4: $H$ is not a point-wise entropy if defined that way - looks like the standard entropy (but $\lambda_i$ does not define a probability distribution)
- "We further check" (p. 7): this paragraph is not self-contained and cannot be understood without reading the appendix

**Strength And Weaknesses:**


Strengths:
- Learning the alignement allows to improve results of already successful models such as ColBERT.
- Few-shot learning based on selecting the right alignment improve the results
- Interesting sparsity regularizer with a target ratio (or number? This is not really clear) of tokens kept for the index

Weaknesses:
- The alignement is not learned
- The sparsity is not really learned since during indexing the ratio is different ($\beta$ vs $\alpha$) and more importantly it is fixed - why this discrepancy?
- Novelty is low (pruning has been proposed in ColBERTer)
- The experimental comparisons lack rigor (compared models have 60x times less parameters, ColBERTer is not included in the experiments, some models could have been tested with the alignement adaptation)
- No complexity analysis (comparing it to ColBERTv2 or ColBERTer)


**Summary Of The Paper:**

In IR, this paper generalizes some neural interaction models like e.g. ColBERT, ME-BERT or COIL, by "learning" the alignement, as well as pruning the multi-vector representations. The model is shown to have a few-sample adaptability on the BEIR dataset.


**Summary Of The Review:**

This paper proposes to (1) use different alignment patterns for few-shot learning (2) sparsify the set of vectors used to compute a document. There are some unclear parts and a lack of motivation (in particular for (2)), and experimental results lack some rigor in the comparison. The alignment few-shot learning could have been more interesting and innovative by looking at various patterns (and motivating them better).

---

> ### Author Response · Authors · 2022-11-19
> **Response to Reviewer sjAr (Part 1)**
>
> Thanks to the reviewer for great comments and suggestions and finding our few-shot learning results interesting. We address the reviewer's concerns in the following order.
>
> > The alignment is not learned
>
> We would like to clarify that the sparsity ratio of alignment is pre-determined and not learned. However, our model end-to-end learns the pairwise alignments between query tokens and document tokens without additional alignment supervision. With the sparse pairwise alignment constraint, the model is forced to learn good alignments between query and document tokens in order to predict the query-document relevance score (Section 2). In addition, as discussed in Section 4.4, the learned pairwise alignments are interpretable by humans. Besides, in Figure 4, we demonstrated how changing pairwise alignment sparsity and algorithm affects retrieval performance.
>
> > Novelty is low (pruning has been proposed in ColBERTer)
> > ColBERTer is not included in the experiments
>
> Thanks to the reviewer for the comments. We would like to appreciate the contribution of ColBERTer and acknowledge that ColBERTer is indeed related to our work. For example, token pruning by salience prediction is also proposed in CoLBERTer.
>
> We want to highlight a number of major differences. ColBERTer mainly concerns in-domain retrieval performance and efficiency; while AligneR focuses on studying the generalization of retrieval models and thus is tested on out-of-domain retrieval tasks. The divergence in research focus leads to differences in model design and evaluation.
>
> ColBERTer is a hybrid retrieval model, fusing single-vector retrieval with multi-vector refinement. And the token pruning is mainly applied to multi-vector refinement. As we concern generalization, AligneR takes the multi-vector retrieval approach as ColBERT because previous works suggest that single-vector retrieval (e.g. only using CLS token for retrieval) does not generalize as well. We recognized that multi-vector retrieval is computationally expensive, which prevents multi-vector models from being widely adopted. In contrast, multi-vector refinement is relatively cheap, especially when token embeddings are cached. Therefore, we design our pruning technique to speed up multi-vector retrieval (Section 3.2). We showed in Section 4.3 that the computation cost of multi-vector retrieval can be drastically reduced by up to 94% with minimal performance loss. In addition, we discovered that multi-vector retrievers can be easily adapted to a new task with few examples (Table 3). With alignment adaption, AligneR significantly outperforms dual-encoder based Promptagator in few-shot retrieval (Table 2), which makes AligneR an appealing solution for generalizable retrieval. To the best of our knowledge, the findings of alignment adaptation are novel and not reported in previous work.
>
> Genuinely, our work complements both ColBERT and ColBERTer. Therefore, we believe that our work is novel and our contributions are significant. As suggested by the reviewer, ColBERTer’s performance on MS MARCO is included in Table 2. And more discussion on ColBERTer is included in the related work section and the introduction.
>
> (to be continued)

---

> > ### Author Response · Authors · 2022-11-19
> > **Response to Reviewer sjAr (Part 2)**
> >
> > > The sparsity is not really learned since during indexing the ratio is different ($\alpha$ vs $\beta$) and more importantly it is fixed - why this discrepancy?
> >
> > > Also, why using two pruning ratios (for training and inference)? How were those ratios determined?
> >
> > These are great questions. We would like to clarify that the sparsity ratio $\alpha$ is pre-determined and not learned. In fact, the sparsity ratio $\alpha$ plays a similar role as the coefficient used in L1 regularization (more on L1 regularization method in the context). Similar to L1 regularization, our proposed entropy regularized linear programming returns values between [0, 1] instead of binary values. In fact, both $\alpha$ and \epsilon (Equation 4) affects the sparsity of the output. The salience features are indeed learned end-to-end by the model which indicates the importance of the tokens.
> >
> > In Section 3.2, we mentioned that the sparsity ratio $\alpha$ is used in both training and refinement and $\beta$ (< $\alpha$) is used in multi-vector retrieval. This design choice of  $\alpha$ vs $\beta$ being different is due to our research focus mentioned above. We realized that multi-vector retrieval is the bottleneck and is much more expensive than multi-vector refinement (as token embeddings are cached for refinement). To improve multi-vector retrieval efficiency, we proposed to prune tokens using a smaller sparsity ratio $\beta$. Since improving multi-vector refinement efficiency is a non-goal, AligneR does NOT prune tokens for multi-vector refinement and $\alpha$ is used for refinement (Section 3.2). As we concern out-of-domain retrieval performance, the optimal sparsity ratio for one task is probably not optimal for another task. Therefore, we only train one AligneR model and change $\beta$ for the multi-vector retrieval stage. Interestingly, as shown in Figure 7, pruning for multi-vector retrieval can even improve retrieval performance for tasks, such as Touche-2020, TREC-covid, climate-fever, etc.
> >
> > As people may expect, pruning for multi-vector refinement could hurt retrieval performance. To illustrate, we trained and evaluated several AligneR models with $\alpha^d = \beta^d$. It is observed that training w/ $\alpha^d = \beta^d$ leads to the worse retrieval performance mainly due to the worse refinement performance caused by token.
> >
> > |                                              | $\beta^d=40$ | $\beta^d=30$ | $\beta^d=20$ | $\beta^d=10$ |
> > |----------------------------------------------|--------------|--------------|--------------|--------------|
> > | Train w/ $\alpha^q=50\%$, $\alpha^d=40\%$    | 38.01        | 38.02        | 37.99        | 37.55        |
> > | Train w/ $\alpha^q=50\%$, $\alpha^d=\beta^d$ | 37.95        | 37.48        | 36.39        | 34.84        |
> >
> > Lastly, we acknowledge that pruning for multi-vector refinement (as proposed in ColBERTer) is a good research direction for zero-shot and few-shot retrieval. Due to the page limit, we won’t be able to study this extensively in this work. We encourage more future research on multi-vector retrievers.
> >
> > > why not using a L1 loss (like in ColBERTer)? The authors should discuss and motivate much more this aspect since it could interesting for other works
> >
> > We thank the reviewer for the great suggestion. We took the advice and conducted the experiment of using L1 regularization for salience prediction. The coefficient of L1 regularization is tuned so that it results in a ballpark number of sparsity on average. The performance of our L1 regularization model is comparable to our proposed entropy-regularized linear programming. The entropy-regularized linear programming is a bit better when $\beta$^d is > 10%. The results are included in Figure 6.
> >
> > One of the main motivations of using entropy regularization is due to its strong controllability over sparsity. In real-world applications, we would like to control the index size for each document in a corpus. It is easy to control sparsity using entropy regularization linear programming. On the other hand, it is not trivial to figure out the coefficient for L1 regularization given a target sparsity ratio. Besides, with a fixed coefficient, L1 regularization results in different sparsity for different documents, making controlling the index size more challenging.
> >
> > We have included more discussion in Section 3.2 to better motivate our proposed method.
> >
> > (to be continued)

---

> > > ### Author Response · Authors · 2022-11-19
> > > **Response to Reviewer sjAr (Part 3)**
> > >
> > > > No complexity analysis (comparing it to ColBERTv2 or ColBERTer)
> > >
> > > ColBERT motivates our model. In Figure 1, we provided a unified formulation to compare various retrieval models, including DPR and ColBERT. The major modeling difference between AligneR and ColBERT is the factorization of the alignment matrix (Figure 2). In addition, the training complexity is compared in Table 1. ColBERTv2 is trained with distillation and model-based hard negative mining. AligneR is in fact trained with a simple recipe, i.e. MS MARCO + fixed hard negatives, as our focus is to demonstrate the effect of alignment adaptation for few-shot retrieval and salience pruning rather than optimizing the scores on the benchmarks.
> > >
> > > Compared to ColBERTer, AligneR modeling and training is simpler than ColBERTer, as we don’t model CLS tokens and we don’t use whole word embeddings. Vanilla whole word embeddings might not work well for out-of-domain retrieval. For example, sentence-piece tokenizer might be a better choice when modeling long words, such as *Isavuconazonium Sulfate*. Such words appear frequently in the biomedical domain.
> > >
> > > > The experimental comparisons lack rigor (compared models have 60x times less parameters, ColBERTer is not included in the experiments)
> > >
> > > > Some extra-care should be taken to compare comparable models (in terms of parameters at least, a 60x ratio is not acceptable)
> > >
> > > This is a great suggestion. Originally, we reported the performance of AligneR_base in Figure 4 and Appendix. We agree with the reviewer that this is not clear and thus we added AligneR_base’s performance on MS MARCO and BEIR in Table 2. Without alignment adaptation, AligneR_base is not as good as ColBERTv2 mainly due to our simpler training recipe (Table 1). We expect that using distillation and model-based hard negatives can further improve AligneR. As mentioned above, we focus on studying few-shot retrieval and we showed that AligneR can be quickly adapted to a new task with a couple of examples via alignment adaptation.
> > >
> > > > some models could have been tested with the alignment adaptation
> > >
> > > Thanks for the suggestion. We have trained 5 additional models with different alignment settings. The results can be found in Figure 4. It is observed that models trained with more number of alignments performed worse than those trained with less number of alignments. Alignment adaptation is effective for all the models. Moreover, it closes the performance gap among different models.
> > >
> > > > Foonote 4:  $H$ is not a point-wise entropy if defined that way - looks like the standard entropy (but  does not define a probability distribution)
> > >
> > > We thank the reviewer for pointing out the issue. Yes, $\lambda$ is not a probability distribution. We used an extension of the entropy function that is applied to any positive vector $\lambda$, following: Csiszár I. On Iterative Algorithms with an Information Geometry Background. 2008. This comment is included in the manuscript for clarity.
> > >
> > > > "We further check" (p. 7): this paragraph is not self-contained and cannot be understood without reading the appendix
> > >
> > > Thanks for the comment. We updated our manuscript to make this more clear to the reader.

---

> > > > ### Comment · Reviewer_sjAr · 2022-11-25
> > > > **Response**
> > > >
> > > > I have some reserves about table 2 (experimental results) since we would have a much stronger argument comparing models with the same number of parameters and state-of-the-art training techniques; there is always a doubt about whether the improvement is cumulative or not.
> > > >
> > > > I however raise my rating to 6 - since the author's response address most of my concerns.

---

> > > > > ### Author Response · Authors · 2022-11-29
> > > > > **Thank you for the feedback**
> > > > >
> > > > > We are glad that our responses have addressed most of your concerns. We appreciate that you raised the score.

---

> ### Author Response · Authors · 2022-11-23
> **Looking forward to your feedback**
>
> Dear reviewer,
>
> We hope that you've had a chance to read our responses. We would really appreciate a reply as to whether our responses and clarifications have addressed the issues raised in the review, or whether there is anything else we can address.

---

### Official Review · Reviewer_waGp · 2022-10-25

**Confidence:** 4
**Correctness:** 3
**Technical Novelty And Significance:** 3
**Empirical Novelty And Significance:** 3
**Recommendation:** 6

**Clarity, Quality, Novelty And Reproducibility:**

What's the retrieval speed and index space? How the hyper parameters influence the speed and space?

**Strength And Weaknesses:**

Strength
1. The authors proposed a new way to learn sparsified pairwise alignments between query and document tokens.
2. The unary saliences can significantly reduce the document token representations minimal performance loss.
3. The method can achieve good performance on BEIR benchmark under the zero-shot setting.

Weakness:
1. No analysis on retrieval speed and index space.
2. The size of the datasets seem not large enough. Most of the datasets have been pre-processed by some existing IR tools. It would be great if the authors could build index from scratch and compare to the method of "Dense Passage Retrieval for Open-Domain Question Answering".


**Summary Of The Paper:**

The authors cast the multi-vector retrieval problem as sparse alignment between query and document tokens. They propose ALIGNER, a novel multi-vector retrieval model that learns sparsified pairwise alignments between query and document tokens (e.g. ‘dog’ vs. ‘puppy’) and per-token unary saliences reflecting their relative importance for retrieval. They show that controlling the sparsity of pairwise token alignments often brings significant performance gains. In a zero-shot setting, ALIGNER scores 51.1 points nDCG@10, achieving a new retriever-only state-of-the-art on 13 tasks in the BEIR benchmark.

**Summary Of The Review:**

I think this is an interesting paper by introducing a new method of sparsified pairwise alignments, and achieve solid performance on benchmark IR dataset. It would be better if the author can discuss more details about the cost on speed/space and further compare with the setting in dense passage retrieval work.

---

> ### Author Response · Authors · 2022-11-19
> **Response to Reviewer waGp**
>
> Thanks to the reviewer for finding our paper interesting and our empirical results solid. We address the reviewer's concerns in the following order.
>
> > No analysis on retrieval speed and index space.
>
> We agree with the reviewer that retrieval speed and index space are one of the major blockers for the multi-vector retriever to be widely adopted. This is one of our major goals in the paper. As mentioned in Section 3.2, we reduce the number of index searches and index space by pruning query tokens and document tokens respectively. We propose unary salience, which indicates token importance and is learned by the model. Only top document tokens are used to build the index, and top query tokens are used to retrieve nearest neighbors from the index. Note that this filtering linearly reduces the index search queries and index space.
>
> In Figure 6, we showed that at retrieval stage, AligneR can prune 70% of query tokens and 80% of document tokens with minimal impact on retrieval performance. This indicates an 80% reduction in the index size and up to a 94% reduction in the computation cost (assuming brute force retrieval). We also included more discussion on retrieval speed and index space in Section 4.3.
>
> In practice, retrieval latency also depends on the implementation details, such as the indexing algorithm, number of parallelism/sharding, floating-point formats, etc. In this work, we focus on the research question of how to make multi-vector retrieval more efficient by salience pruning.
>
> > The size of the datasets seem not large enough. Most of the datasets have been pre-processed by some existing IR tools. It would be great if the authors could build index from scratch and compare to the method of "Dense Passage Retrieval for Open-Domain Question Answering".
>
> We would like to clarify that none of our datasets are preprocessed by other IR tools, and we build the index from scratch, as mentioned in Section 2.1 and 4.1. Specifically, we pre-compute all the document token embeddings for each corpus that we evaluate on, and build index on these vectors. Besides, we evaluate our model on MS MARCO and BEIR benchmarks, which are commonly used datasets by the research community. Many of these datasets have large corpora. MS MARCO has 8.8 million documents with average length 56, FEVER has 5.4 million documents with average length 85, and HotpotQA has 5.2 million documents with average length 46 (more statistics are available in https://arxiv.org/pdf/2104.08663.pdf). Our experiments validate the accuracy and efficiency of AligneR on these datasets (Table 2).
>
> As suggested by the reviewer, we add the results of DPR (Dense Passage Retrieval for Open-Domain Question Answering) to our paper (see Table 2). Please note that we have already reported the results of GTR, which is also a dual encoder retriever and improves upon DPR with advanced training strategies. The DPR results are worse than the other baselines, such as GTR and ColBERTv2.
> (DPR results, and some other models, are available in the BEIR paper. Maybe expand Table 2 with more results.)

---

> ### Author Response · Authors · 2022-11-23
> **Looking forward to your feedback**
>
> Dear reviewer,
>
> We hope that you've had a chance to read our responses. We would really appreciate a reply as to whether our responses and clarifications have addressed the issues raised in the review, or whether there is anything else we can address.

---

### Author Response · Authors · 2022-11-19
**General Response**

We thank all reviewers for their thoughtful comments. We have incorporated the suggestions in the latest version of our manuscript. More specifically, we

- Added AligneR_base, GTR_base (https://arxiv.org/abs/2112.07899) and DPR (https://arxiv.org/abs/2004.04906) to Table 2 and reported their performance on MSMARCO, zero-shot and few-shot performance on the BEIR benchmark for better comparison.  Originally, the performance of AligneR_base is reported in Figure 3 and in Appendix.
- Conducted the baseline experiment of using L1 regularization for sparse salience prediction, as suggested by (Reviewer sjAr). The results are reported and discussed in Figure 6 and Section 4.3.
- Added more discussion in Section 3.2 to motivate using entropy regularized linear programming for unary salience prediction.
- Trained 5 additional AligneR models with different pairwise alignment settings to demonstrate the effectiveness of few-shot alignment adaptation. The results are reported and discussed in Figure 4 and Section 4.2.

We address each reviewer’s comments separately below.

---

### Author Response · Authors · 2022-11-23
**Looking forward to your feedback**

Dear reviewers,

We hope that you've had a chance to read our responses. We would really appreciate a reply as to whether our responses and clarifications have addressed the issues raised in the review, or whether there is anything else we can address.

---

### Decision · Program_Chairs · 2023-01-20

**Decision:**

Reject

**Justification For Why Not Higher Score:**

There are three weaknesses: no complexity analysis, low rigor in experimentation and the paper being difficult to understand (see writeup for details).

Additionally, while all reviewers agreed that the paper is marginally above the acceptance threshold, none of the reviewers is particularly excited about the paper.

**Justification For Why Not Lower Score:**

N/A

**Metareview: Summary, Strengths And Weaknesses:**

This paper discusses multi-vector document retrieval based on embeddings as a way of performing Information Retrieval. The paper introduces a sparse alignment perspective that offers a unifying view of previous dense retrieval schemes. The paper proposes Aligner, a model that uses sparse alignment between terms combined with unary weights for term salience.

Experimental evaluation shows that the model outperforms other dense retrieval techniques. Further, the paper shows that using the unary salience score, the number of terms can be pruned to improve storage and runtime requirements with minimal losses of accuracy.

The paper has several weaknesses:
- There is no analysis of runtime complexity. The paper contains study showing that terms can be pruned and runtime improved, but it is unclear what is the starting point and how it compares to other algorithms. In fact, it is not completely clear from the paper how fast sparse methods can be used to limit the number of candidates for which the full alignment is computed.
- The experimental comparisons lack rigor -- models of different sizes and using different training data are compared. Since models from previous work are categorized in the same framework, it should be easy to compare models by changing a single dimension to establish the correct baselines and understand what makes the method work.
- It is hard to understand the paper and the different technical details. This is epitomized by all of the reviewers having different fundamental misunderstandings about the paper that the authors had to correct in the response period.